Manuscript prepared for Earth Syst. Dynam.
with version 2014/09/16 7.15 Copernicus papers of the LaTeX class copernicus.cls.
Date: 26 April 2018

# Estimating sowing and harvest dates based on the Asian Summer Monsoon

Camilla Mathison[1], Chetan Deva[2], Pete Falloon[1], and Andrew J Challinor[2]

[1]Met Office, FitzRoy Road, Exeter, EX1 3PB, UK
[2]School of Earth and Environment, Institute for Climate and Atmospheric Science, University of Leeds, Leeds, LS2 9AT, UK

*Correspondence to:* Camilla Mathison (camilla.mathison@metoffice.gov.uk)

**Abstract.** Sowing and harvest dates are a significant source of uncertainty within crop models especially for regions where high-resolution data are unavailable or, as is the case in future climate runs, where no data are available at all. Global datasets are not always able to distinguish when wheat is grown in tropical and sub-tropical regions, they are also often coarse in resolution. South Asia is
one such region where large spatial variation means higher resolution datasets are needed, together with greater clarity for the timing of the main wheat growing season. Agriculture in South Asia is closely associated with the dominating climatological phenomena, the Asian Summer Monsoon (ASM). Rice and wheat are two highly important crops for the region, rice being mainly cultivated in the wet season during the summer monsoon months and wheat during the dry winter. We present
a method for estimating the crop sowing and harvest dates, for rice and wheat, using the ASM onset and retreat. The aim of this method is to provide a more accurate alternative to the global datasets of cropping calendars than is currently available and generate more representative inputs for climate impact assessments.

    We first demonstrate that there is skill in the model prediction of monsoon onset and retreat for
two downscaled General Circulation Models (GCMs) by comparing modelled precipitation with observations. We then calculate and apply sowing and harvest rules for rice and wheat for each simulation to climatological estimates of the monsoon onset and retreat for a present day period. We show that this method reproduces the present day sowing and harvest dates for most parts of India. Application of the method to two future simulations demonstrates that the estimated sowing
and harvest dates are successfully modified to ensure that the growing season remains consistent with the internal model climate. The study therefore provides a useful way of modelling potential growing season adaptations to changes in future climate.

## 1 Introduction

Field studies dominate the modelling literature on crops and agriculture. Many crop models are developed and applied at the field scale using site specific observations to drive models and optimize outputs. The growing awareness of climate change and the likely impact this will have on food production has generated a demand for regional and global assessments of climate impacts on food security through for example, projects such as Agricultural Model Intercomparison and Improvement Project (AgMIP-Rivington and Koo (2010); Rosenzweig et al. (2013, 2014)), the Inter-Sectoral Impact Model Intercomparison Project (ISIMIP-Warszawski et al. (2013, 2014)) and Global Gridded Crop Model Inter-comparison (GGCMI-Elliott et al. (2015)). Recent work in such climate-crop impact studies has sought to quantify uncertainty from the quality and scale of input data. A result from this work is that for global scale simulations, planting dates are a significant source of uncertainty (Frieler et al., 2016; Elliott et al., 2015).

Aside from their use in modelling studies, deciding when to plant crops is a significant challenge particularly in water scarce regions such as parts of Sub-Saharan Africa (SSA - Waongo et al. 2014), South and South East Asia (Kotera et al., 2014). These regions have crop sowing dates that are closely associated with the onset of the rainy season. Any prolonged dry spells of more than 2 weeks after sowing could have serious consequences leading to crop failure or significant yield reduction because top soil layers dry out preventing germination (Laux et al., 2008). For large parts of SSA deciding when to sow determines the length of the crop duration for the agricultural season and is therefore an important tactical decision (Waongo et al., 2014).

Planting dates can be determined using a number of different methods, for example, Kotera et al. (2014) propose a cropping calendar model for rice cultivation in the Vietnam Mekong Delta (VMD). The Kotera et al. (2014) model estimates the sowing date based on the the suitability of the land for crops given any flooding, salt water intrusion or erratic monsoon rains; these are important factors for the water resources of the VMD region. Alternatively Laux et al. (2008, 2010) use a fuzzy logic-based algorithm developed to estimate the onset of the rainy season in order to examine the impact of the planting date for the SSA. In the General Large Area Model (GLAM-Challinor et al. (2004a)), the sowing date can be estimated by the model based on the soil moisture conditions, with the crop sown when surface soil moisture exceeds a specified threshold during a given time window and crop emergence occurring a specified time after sowing. Waha et al. (2012) base their estimates of sowing dates at the global scale on climatic conditions and crop specific temperature thresholds, therefore providing a suitable method for taking climate change into account. However the Waha et al. (2012) method is not really intended for use in irrigated multiple cropping regions. Elliott et al. (2015) describe how sowing dates are defined in the GGCMI project. The GGCMI protocols use a combination of Sacks et al. (2010), Portmann et al. (2010) and model data to define sowing dates, thus highlighting the challenges in defining a complete, accurate dataset of sowing and harvest dates. This has influenced and driven the development and application of crop models on broader

scales. In this study we are considering the whole South Asia region, this is a large scale problem with complicated cropping patterns, which means that assumptions and generalizations need to be made across a region with a wide variety of climatic conditions and cropping environments (soils etc). Waha et al. (2013) highlight that global crop calendars such as those used in the GGCMI often only report individual crops, therefore limiting their usefulness for regions with multiple cropping
systems.

The growing interest in climate change and food security has influenced the development of crop models for use in future climate impact assessments (Frieler et al., 2016); this represents a different challenge for crop models in terms of the input data used. ISIMIP simulations use time varying crop management data until 2005 after which the data are held fixed at 2005 levels for the remainder
of the simulations (Frieler et al., 2016). Fixing crop management to present day practices is not really suitable for adaptation studies (van Bussel et al., 2015). The assumption that there will be no large shifts in climate causing sowing and harvest dates to change significantly from the present day, could lead to the sowing and harvesting of crops in the model in the future at unrealistic times of the year. Thus the appropriate sowing and harvest dates used in future simulations depends on
the intended application for the simulations. In many adaptation studies, impacts without adaptation are assessed using present day estimates of sowing dates, then the sowing dates are adjusted in response to climate change to assess the benefits of adaptation (Lobell, 2014). Challinor et al. (2017) suggest using autonomous adaptation in simulations in order to avoid overestimating the effects of adaptation. On this basis there is a requirement for estimates of sowing and harvest dates for climate
simulations that remain consistent with the future model climate. Thus making estimates of sowing and harvest dates important not only for understanding the present day, but also for use in future simulations especially when considering potential adaptation to climate change.

Agriculture in South Asia is dominated by the Asian Summer Monsoon (ASM). Kharif and Rabi are the two main seasons in South Asian agriculture and these correspond to summer and win-
ter/spring growing seasons respectively. Rice-Wheat systems are a major crop rotation across South Asia. Kharif crops include rice which is usually sown during the monsoon, and harvested in the autumn. Sowing and harvest dates for rice cultivated during the Kharif season vary between states, with rice traditionally sown in some locations with the first rains of the monsoon, while other regions such as eastern parts of the Indo Gangetic Plain (IGP) tend to plant rice late into June when
the monsoon is fully established (Erenstein and Laxmi, 2008). Rabi crops include wheat which is mainly cultivated during the dry season (Erenstein and Laxmi, 2008; Singh et al., 2014). The close association of the sowing dates of these crops and the ASM offer the potential for a new method of defining the cropping calendar for this important rotation.

Rice-wheat systems, particularly those in Pakistan (Erenstein et al., 2008) and the Indo Gangetic
Plain (IGP), tend to plant varieties like Basmati that take a long time to mature (Erenstein and Laxmi, 2008). Since this delays wheat planting, this has a direct impact on wheat yield. In the Eastern IGP

this is a particular problem as the season for which wheat is viable is relatively short (Erenstein and Laxmi, 2008; Laik et al., 2014; Jat et al., 2014). Any delay between the rice harvest and wheat planting can have a large impact on the success of the wheat crop as this will reduce the time available
before the temperatures get too high for the successful cultivation of wheat (Joshi et al., 2007). The time between the rice harvest and wheat sowing also depends on the time it takes to ensure the soil is in a suitable condition for wheat sowing after the rice harvest. Erenstein and Laxmi (2008) describe the zero-tillage approach which allows for a reduced turn-around time between the harvest of rice and sowing of wheat. Potential avenues by which the uncertainty from sowing and harvest dates can
be reduced in inputs to crop simulations include:

- The use of higher resolution regional data sets of recorded sowing and harvest dates for crop calendars rather than existing global data sets.

- The use of new methods for estimating crop calendars in the absence of higher resolution regional data sets.

**1.1 Motivation**

The correct representation of the crop duration within crop models are crucial for the interpretation of the important outputs from the model. For example if the datasets used for sowing and harvest dates are inaccurate, the simulations could grow crops during the wrong season, thereby affecting the reliability of the simulated water use and crop yield. The main differences between the regional
Bodh et al. (2015) dataset and the global Sacks et al. (2010) data are for spring wheat. Spring wheat grown in winter is misclassified as winter wheat in the Sacks et al. (2010) data. This is discussed by Sacks et al. (2010) as a potential limitation when using the data for tropical and subtropical regions. Spring wheat is the more common type of wheat grown in the South Asia region (Hodson and White, 2007) because minimum temperatures there are not low enough to allow vernalization to take place,
which is needed for winter varieties of wheat (Sacks et al., 2010; Yan et al., 2015).

Figure 1 shows the averaged rice (green rectangles) and wheat (orange rectangles) growing season durations for Sacks et al. (2010) (diagonal hatching) and the Bodh et al. (2015) dataset (perpendicular hatching-labeled MinAg) over-laid on the present day South Asia averaged precipitation climatology and estimates of the monsoon onset and retreat. This illustrates the differences between
the Bodh et al. (2015) and Sacks et al. (2010) datasets showing that in Sacks et al. (2010), the main growing period for both rice and wheat appears to be during the monsoon. While rice is usually grown during the monsoon it is not typical that wheat should be grown during this period for this region. The growing season durations for the Bodh et al. (2015) dataset (See Fig. 1 - perpendicular hatching rectangles labeled MinAg) are more typical of this region with rice (green) growing during
the monsoon and wheat (orange) growing during the dry season. Figure 1 highlights that where a global dataset is unable to establish exactly when wheat is grown in tropical regions, an alternative is needed.

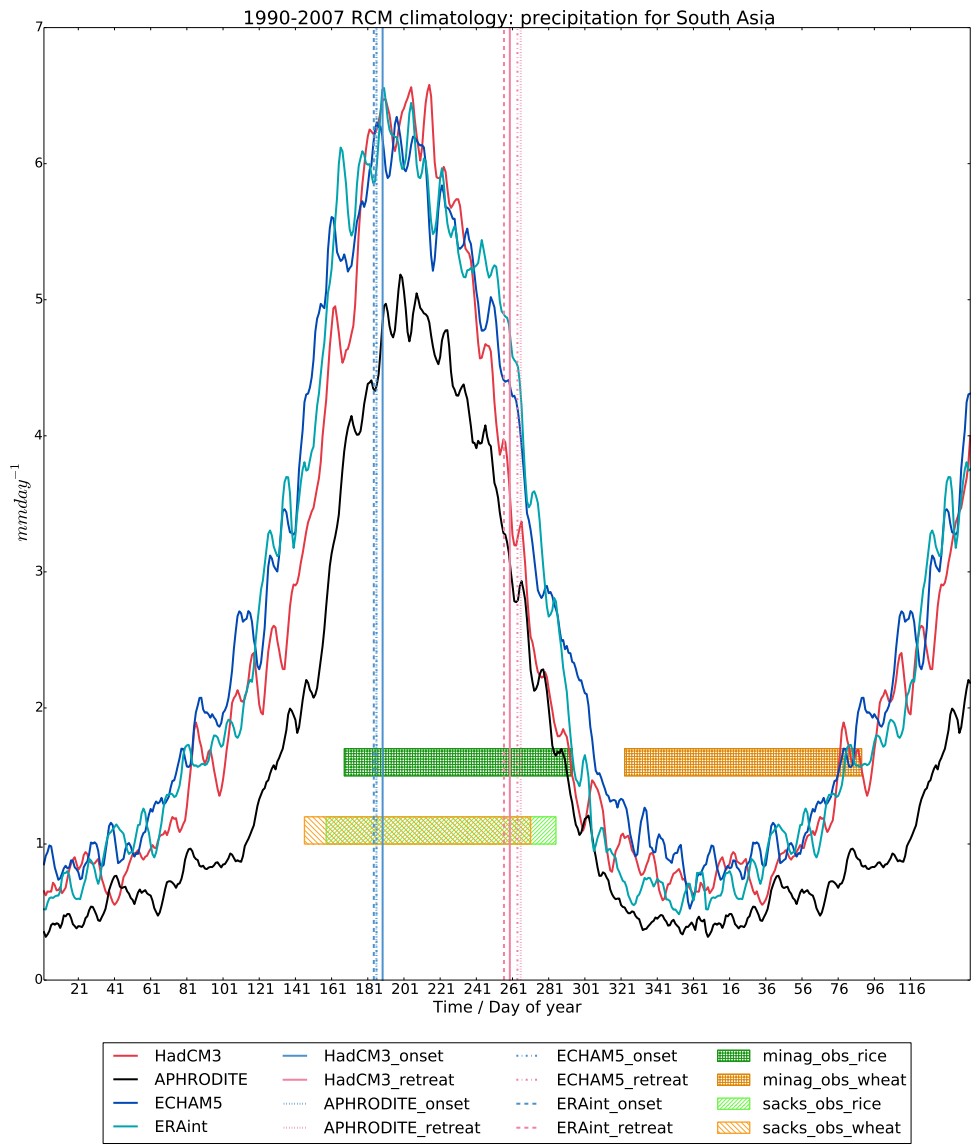

**Figure 1.** The one and a half year precipitation climatology for the 1990-2007 period averaged for South Asia for each simulation (ERAint-cyan line, ECHAM5-blue line, HadCM3-red line) and APHRODITE observations (black line) using a 5-day smoothed rolling mean. Also shown are the growing seasons also averaged for 1990-2007 for South Asia for wheat (orange) and rice (green) from two datasets; Sacks et al. (2010) (diagonal hatching -labeled sacks) and Bodh et al. (2015) (perpendicular hatching-labeled minag) and the monsoon onset (blue vertical lines) and retreat (pink vertical lines) from each of the simulations (APHRODITE-dotted, ERAint-dashed, HadCM3-solid,ECHAM5-dash dot).

Crop models such as those described by Challinor et al. (2003, 2004b) and Osborne et al. (2014) require sowing information such as a sowing date or a sowing window, with the crop model integrating an effective temperature over time as the crop develops. The effective temperature is a

function of air or leaf temperature and differs between models. The integrated effective temperature in each development stage is referred to as the thermal time of that development stage (Cannell and Smith, 1983; McMaster and Wilhelm, 1997) (there may also be an additional photoperiod length dependence). The thermal time in each development stage is typically set by the user, and can be calibrated to simulate different varietal properties. Where these varietal properties are unavailable e.g. for the global analysis in Osborne et al. (2014), in order to mimic the spatial variation in the choice of crop variety, these thermal times were determined from sowing and harvest dates and the temperature climatology which allowed them to vary spatially. This ensures that during the simulation, the crop develops over the course of the crop season starting at the sowing date and ending at approximately the harvest date (i.e. the harvest date is the average over the course of climatological period used). The use of this predefined thermal time ancillary drives the requirement for providing both a sowing and harvest date. Reliable high resolution datasets for sowing and harvest dates are often unavailable for either the region or the time period that is needed. In addition there is a demand for sowing and harvest dates that maintain consistency with the model climate. Therefore, in this paper we propose a new method, outlined in Fig 2, for estimating sowing and harvest dates for use in the large-scale modelling of the rice-wheat rotation in South Asia using estimates of monsoon onset and retreat. This method does not require large amounts of data and the user can elect to use either the sowing input data or if needed, both sowing and harvest data to run their chosen crop model. The main objectives of this study are:

– To develop a method for determining sowing and harvest dates for modelling the rice-wheat rotation in South Asia based on the ASM.

– To test the method in current and future climates.

We therefore present the methodology in Sect. 2. We show the proposed method is viable and show it works in Sect. 3. Discussion of the results and conclusions are provided in Sect. 4 and Sect. 5 respectively.

## 2 Methodology

The methodology is summarized in the flow chart in Fig. 2. The model datasets, described in detail in Sect. A of the Appendix, include General Circulation Models (GCMs) and a Regional Climate Model (RCM). GCMs provide spatially consistent boundary data to an RCM, which generates 25km regional fields (see Fig.2 blue boxes). The two GCMs used in this analysis were specifically selected because they were able to capture main features of the ASM (See Sect. A of the Appendix). RCMs are based on the same physical equations as GCMs and therefore represent the entire climate system including the carbon and water cycle. Their higher resolution allows a better representation of the regional-scale processes adding detail to fields like precipitation (Mathison et al., 2015). The

individual RCM simulations (also called HNRCMS - see Appendix Sect. A) used in this analysis
are referred to using their global driving data abbreviations; HadCM3, ECHAM5 and ERAint as
described in Appendix Sect. A. Precipitation fields are used to generate a precipitation climatology
which are used to calculate monsoon statistics (See Sect. 2.2) from which sowing and harvest dates
are estimated; shown by the pink rectangles (see Sect. 2.3). These estimated sowing and harvest
dates are referred to as relative monsoon sowing and harvest dates (see Fig. 2). Observations are
used throughout the process to ensure the method is viable and produces sensible results, these are
described in Sect. 2.1 and shown by the green boxes.

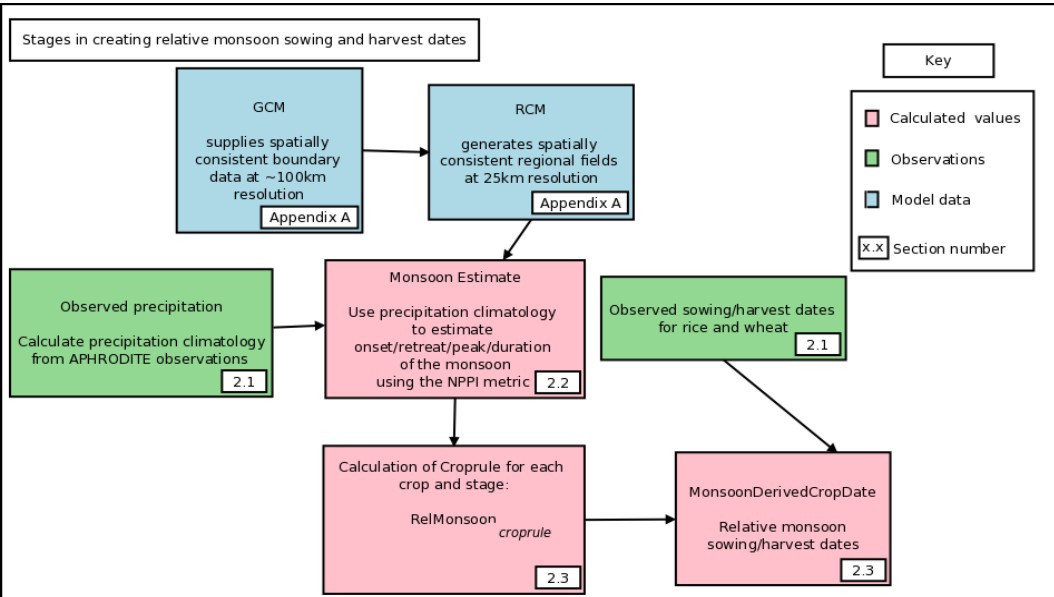

**Figure 2.** A flow chart summarizing the methodology. The blue rectangles represent datasets that are used
within the methodology, green rectangles represent observations and pink rectangles represent any calculations
parts of the methodology.

## 2.1  Observations

In order to demonstrate the viability of the methodology outlined in Fig. 2 we compare the simulated
precipitation with observations from the Asian Precipitation-Highly Resolved Observational Data
Integration Towards the Evaluation of Water Resources (APHRODITE –Yatagai et al., 2012) dataset
in Section 2.2.1. APHRODITE is a daily, 0.25° resolution land only gridded dataset that is also used
in Mathison et al. (2015) to show that the RCMs in this analysis capture the general hydrology of
the region. The monsoon is a highly variable and complex phenomenon that currently not all climate
models are able to represent; this may mean that some climate models would not yet be suitable for
using with this method, which relies on a good representation of the monsoon. The method presented
in Fig. 2 will become more robust with improving representations of the monsoon in climate models.

The datasets used for sowing and harvest dates include a global dataset, Sacks et al. (2010) and a regional dataset, Bodh et al. (2015) from the Government of India, Ministry of Agriculture & Farmers welfare. The Bodh et al. (2015) data is referred to from here on as MinAg data. The MinAg observations of sowing and harvest dates for rice and wheat are given as a range of days of year. The midpoints of these observed ranges are calculated and compared against the midpoints of the model pentads for onset and retreat in day of year. As a post-processing step the differences are then masked using crop areas from the International Crops Research Institute For The Semi-arid Tropics (ICRISAT, 2015) so that only the areas where rice or wheat are grown are considered.

## 2.2 Estimating monsoon onset and retreat

There are a wide variety of metrics for estimating the monsoon onset and retreat. Some are specific to agriculture and include representation of breaks in the monsoon (Moron and Robertson, 2014). More general metrics include a combination of meteorological variables such as 850hPa wind and precipitation as in Martin et al. (2000), or only use precipitation, such as in Sperber et al. (2013) and the Normalised Pentad Precipitation Index (NPPI) (Lucas-Picher et al., 2011). The NPPI and Sperber et al. (2013) methods both use a long term climatological average of precipitation because the model data are too noisy to calculate the monsoon statistics per year. Agricultural specific definitions of monsoon onset and retreat represent breaks in the monsoon which can adversely affect the germination of crops. However these metrics are not as effective when used in conjunction with long term average precipitation fields such as those used here. This is probably because the breaks that occur in the monsoon are quite variable from year to year and are smoothed out within the climatology. The approach by Sperber et al. (2013) defines monsoon onset as the pentad where the relative rainfall exceeds 5 mm day$^{-1}$ during the May-September period. However, Sperber et al. (2013) re-grid to the GPCP rainfall dataset (Huffman et al., 2001) which is much coarser resolution than the APHRODITE data used here. The NPPI metric uses Eq.(1) to estimate monsoon onset, retreat, peak and duration.

$$NPPI = \frac{P - P_{min}}{P_{max} - P_{min}} \tag{1}$$

where $P$ is the unsmoothed pentad precipitation climatology and $P_{min}$ and $P_{max}$ are the annual minimum and maximum at each gridbox respectively. The monsoon onset is then defined as the pentad in which the NPPI exceeds 0.618 for the first time and withdrawal as the last time the NPPI drops below this threshold in the year. The NPPI only reaches a value of 1.0 once in the annual cycle which corresponds to the monsoon peak. In the NPPI method the only regridding that takes place is to ensure the model and observations are on the same grid, as they are both 25km resolution there is no loss of resolution in doing this. The threshold for NPPI is also independent of the resolution of the data which is not the case for the Sperber et al. (2013) method. The NPPI metric has been

successfully applied previously by Lucas-Picher et al. (2011) to analyse the monsoon of models of a similar resolution to the simulations used here (See Fig 2). Therefore in this analysis in the same way that Lucas-Picher et al. (2011) uses the 1981-2000 climatology, we use a 1990-2017 climatology. The pentad provided by the NPPI is representative of the climatological period and therefore cannot be compared to a particular year, however the pentad can be used to find the 5-day window for the climatological period where onset and retreat typically occur which can then be compared to APHRODITE observations also averaged for that period. We use the NPPI metric to calculate the pentad of the monsoon onset, retreat, peak and duration for the APHRODITE observations and the three HNRCM simulations.

### 2.2.1 Comparison of model monsoon onset and retreat with precipitation observations

Figure 3 shows plots of the onset (left column) and the retreat (right column) of the South Asian Summer Monsoon as defined using the NPPI described in Sect. 2.2. The NPPI index for the climatology of the APHRODITE precipitation observations (Yatagai et al., 2012) are shown in plots (a) and (b) of Fig. 3 for comparison with the precipitation climatology for each of the HNRCMs shown; ERAint (c and d), HadCM3 (e and f) and ECHAM5 (g and h). The white regions are areas where the threshold was exceeded at the first pentad, this implies the monsoon had already started at the first pentad which suggests a model bias and therefore these regions were masked out. Figure 4 shows the differences between the model onset (retreat) and APHRODITE onset (retreat) for each model. On average the difference between the monsoon onset in APHRODITE and the HNRCM simulations is between 1 and 7 days and the difference between the retreat in APHRODITE and the HNRCM simulations is between 4 and 10 days. However there are regions where the differences between the APHRODITE monsoon statistics are much larger than this, these are highlighted by the darker red and blue regions in Fig. 4. In general for most of India the HNRCMS are within 25 days of the APHRODITE observations, with the regions where the differences are larger explained by different monsoon characteristics, for example the South of India and the Bangladesh region (this is discussed further in Sect. 4.1).

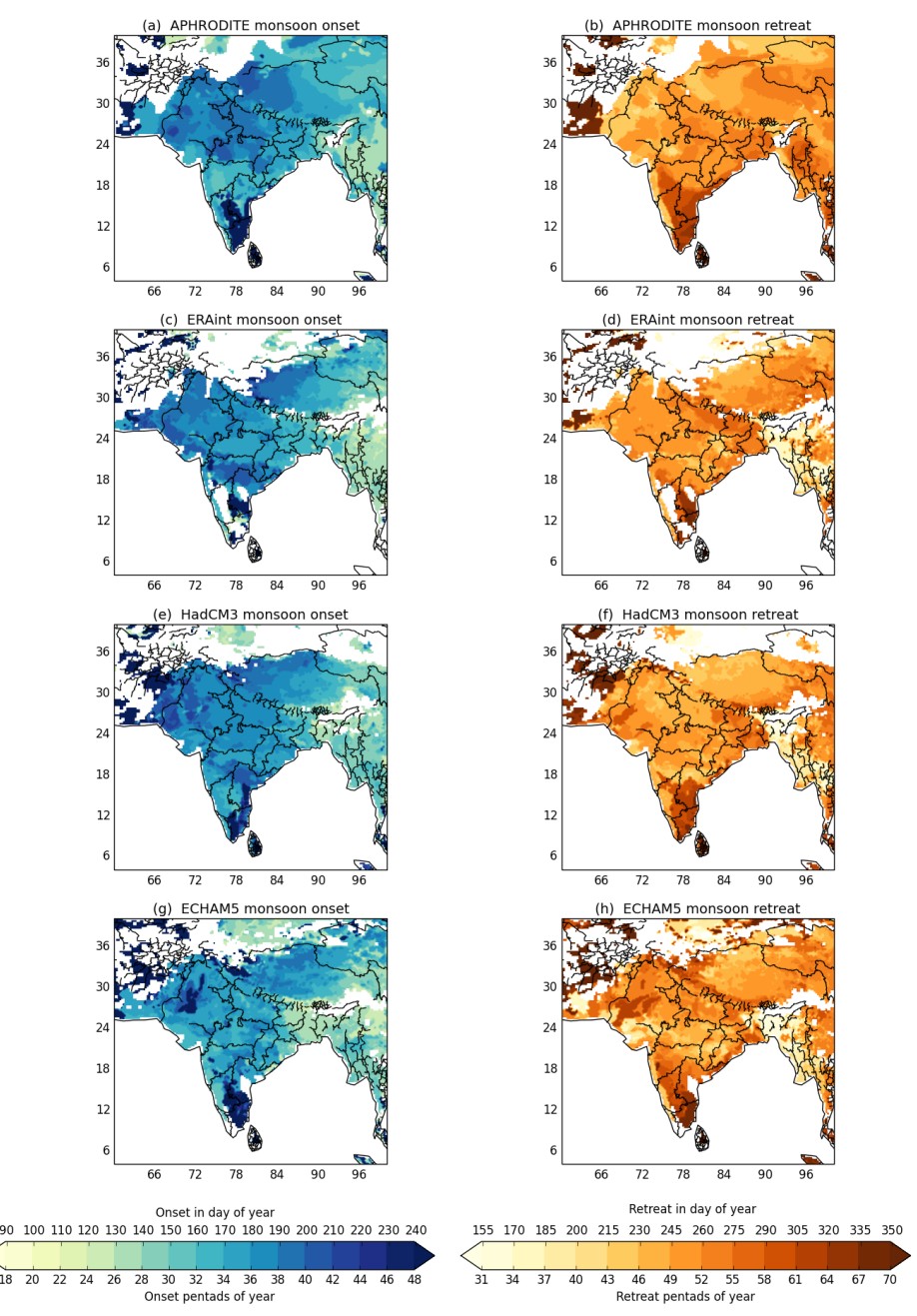

**Figure 3.** Plots of the 1990-2007 monsoon statistics; monsoon onset (left column) and retreat (right column). The APHRODITE precipitation observations (a and b) are shown and the three model simulations; ERAint (c and d), HadCM3 (e and f) and ECHAM5 (g and h) calculated using the NPPI metric. White areas are the regions where the model precipitation exceeds the threshold indicating the start of the monsoon at the initial pentad, this does not imply early monsoon but more likely a model bias in the precipitation at this location.

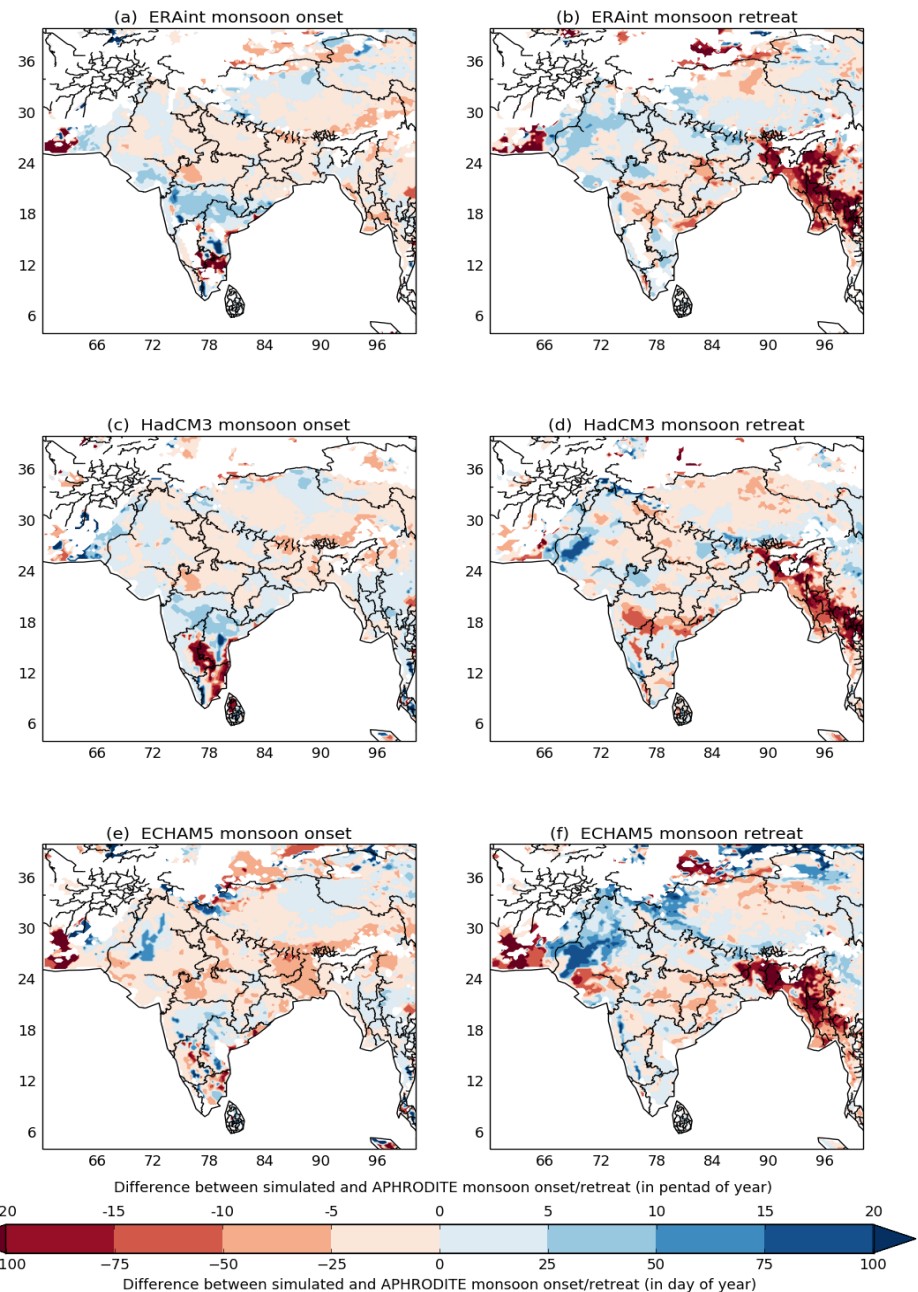

**Figure 4.** Plots of the 1990-2007 difference between model simulations and APHRODITE observations for the monsoon statistics; monsoon onset (left column) and retreat (right column); ERAint (a and b) HadCM3 (c and d) and ECHAM5 (e and f) calculated using the NPPI metric.

### 2.3 Calculating sowing and harvest dates from monsoon characteristics

We use estimates of the monsoon onset and retreat together with present day rules on sowing and harvest for rice and wheat, referred to as crop rules to calculate the sowing and harvest dates relative to the monsoon (See Fig.2). This method allows any crop model that uses, for example a driving dataset similar to APHRODITE or the HNRCMs, to derive sowing and harvest dates that are consistent with the monsoon of the driving data (see Fig.2). Thus growing the crop at the appropriate time of the year i.e rice is kept during the monsoon period and wheat is sown and harvested during the dry season. The monsoon is a highly variable phenomena, however the use of a long term average (climatology) to calculate the monsoon statistics smooths out their large inter-annual variability. This highlights the consistency between the sowing and harvest dates and the monsoon statistics. Therefore we do not expect the monsoon statistics to be exactly the same as the observed sowing and harvest dates. Rather, this method relies on consistency between the climatological estimate of the monsoon statistics and the sowing and harvest dates across the region. The introduction of a croprule then moves the monsoon statistic to more closely reflect the observed sowing and harvest dates. This means that even if the difference between the most relevant monsoon statistic and the observed sowing or harvest date is large then the difference is similar across India. Although these sowing and harvest events may not always be dictated entirely by the monsoon, the phenomena provides the broader seasonality associated with the crop seasons in this region. The consistency between the crop practices and the monsoon statistics across the region, provides the empirical relationship exploited here to estimate the sowing and harvest dates for use in both present day and future crop simulations. These sowing and harvest dates are not really intended to offer advice to farmers on when to sow or harvest on a year to year basis, rather it provides a way for sowing and harvest dates to remain relevant to this major climatological feature. A key assumption is that the monsoon remains a defining feature of the crop seasons for South Asia in the future.

#### 2.3.1 Calculation of monsoon derived estimates of sowing and harvest dates for rice and wheat

We use the precipitation climatologies from APHRODITE precipitation observations and each of the HNRCM simulations (See Fig.2) by calculating the difference between the monsoon onset (or retreat) and the observed MinAg sowing (or harvest) dates for each crop (See Fig.2). These differences are per gridbox. We then calculate a weighted area average (using the Met Office (2018) package) to produce a crop rule for the whole region for each crop and stage; these are listed in Eq. 2. Collectively the crop rules given in Eq. 2 are referred to as $RelMonsoon_{croprule}$. This provides a simple rule that can be applied across the region, even where observations are not available. Although calculating a rule per gridbox would provide excellent results where observations were available, it would

limit the usefulness of the method where observations were not available, which is one of the main aims of this approach.

$$RiceSowingCroprule = AreaAverage(MonsoonOnset - RiceSowing)$$
$$RiceHarvestCroprule = AreaAverage(MonsoonRetreat - RiceHarvest)$$
$$WheatSowingCroprule = AreaAverage(MonsoonRetreat - WheatSowing)$$
$$WheatHarvestCroprule = AreaAverage(MonsoonOnset - WheatHarvest)$$

(2)

The $RelMonsoon_{croprule}$ is then applied to the monsoon onset and retreat field to provide an estimate of sowing and harvest dates for rice and wheat based on the monsoon. We refer to these estimates of sowing and harvest dates as 'monsoon derived crop dates' for brevity.

$$MonsoonDerivedCropDate = MonsoonStatistic - RelMonsoon_{croprule}$$  (3)

where the $MonsoonStatistic$ can be monsoon onset or retreat and the $RelMonsoon_{croprule}$ is
one of the four crop rules given in Eq. 2

The spatial variability of the monsoon derived sowing and harvest dates is accounted for by the monsoon onset and retreat in the climatology used to calculate the $RelMonsoon_{croprule}$. The monsoon derived sowing and harvest dates for both the APHRODITE and HNRCM simulations are provided and compared against MinAg observed sowing and harvest dates in Sect. 3.2. The calcu-
lation of the $RelMonsoon_{croprule}$ is based on observations for India (from MinAg and ICRISAT (2015)) and therefore the analysis for the present day in Sect. 3.2 focuses on these areas. On the basis that most of the South Asia region is dominated by the ASM, the $RelMonsoon_{croprule}$, though tuned using India observations, can be applied to any region dominated by the ASM in order to estimate sowing and harvest dates for larger areas with a rice-wheat rotation (see Sect 3.3). The method
does not currently perform as well for parts of southern India where the climate is influenced by the Northeast monsoon but could be modified to provide better results for these areas. In Sect. 3.2, we compare the monsoon derived estimates of sowing and harvest dates for the period 1990-2007 with the MinAg range of sowing and harvest dates to establish if the method shown in Fig. 2 gives good results. There are four datasets used throughout this analysis; APHRODITE and the three HNRCMS.
Where three of the four datasets provide sowing or harvest dates that are within the MinAg range the method is said to give good results, where two of the four datasets are within the MinAg range the results from the method are said to be fair. If no datasets are within the MinAg range the method is classed as poor. The sowing and harvest dates are presented for each state in Sect. 3.2.

## 2.4 Demonstration using monsoon derived estimates of sowing and harvest dates for two future periods

The method summarised in Fig. 2 is applied to two future periods using the ECHAM5 and HadCM3 RCM simulations (described in Sect. A of the Appendix). Global mean temperatures are used (within the High-End cLimate Impacts and eXtremes project - HELIX) to define the future climate in terms of specific warming levels (SWLs), i.e considering a 2°C, 4°C and 6°C world. The use of time periods is much more common than SWLs, however SWLs enable the analysis to focus less on the climate scenarios and more on what the world will look like at 2°C, 4°C and 6°C (Gohar et al., 2017). This will differ depending on when the threshold is passed. The SWL approach is therefore a benefit as it means that new scenarios that are developed as part of new model intercomparison projects can be compared against older ones from previous projects. Although the older scenarios may not contain the most up-to-date socio-economic information they are no less likely than the newer scenarios. The simulations used here are for the period 1965 to 2100 and therefore only the 2°C threshold for global mean temperature is actually passed during these simulations. For HadCM3 this occurs in 2047 and for ECHAM5, 2055. Therefore the two future periods used in this analysis are 2040-2057 and 2080-2097. The 2040-2057 period is chosen because it includes the year that the global mean temperature exceeds 2°C in the two simulations and the 2080-2097 period is chosen because it is furthest into the future in these simulations and therefore likely to show the greatest warming. The length of the two future analyses periods has been chosen for consistency with the ERAint RCM simulation which is only available for the period 1990-2007. Although the threshold of 2°C is exceeded globally it is important to note that the relationship between the projected global mean change in temperature and the regional climate change in temperature for South Asia is complicated. Heat and moisture and how they vary across the globe are not evenly distributed with land warming faster than the ocean (Christensen et al., 2013), therefore the actual temperature change experienced in South Asia may be higher than the global mean change.

## 3 Results

We compare the model monsoon to the monsoon calculated from precipitation observations to demonstrate that the model is able to reproduce the monsoon (See Sect. 2.2.1) and therefore the methodology summarized in Fig. 2 and Sect. 2 is viable. In Sect.3.1 we compare the simulated monsoon with the observed sowing and harvest dates in order to calculate the monsoon derived sowing and harvest dates and compare these new simulated sowing and harvest dates with the observations. We then show results from applying the method in Sect. 3.2. As a demonstration, we also apply the method to two future periods in Sect.3.3.

### 3.1 Comparing observed sowing and harvest dates with estimates of monsoon onset and retreat

The climatology shown in Fig. 1 shows that on average the observed rice and wheat sowing and harvest dates from MinAg align well with the monsoon onset and retreat in the simulations. Observed rice sowing dates generally compare well with the monsoon onset in the model as shown in Fig. 5 and Fig. 6.

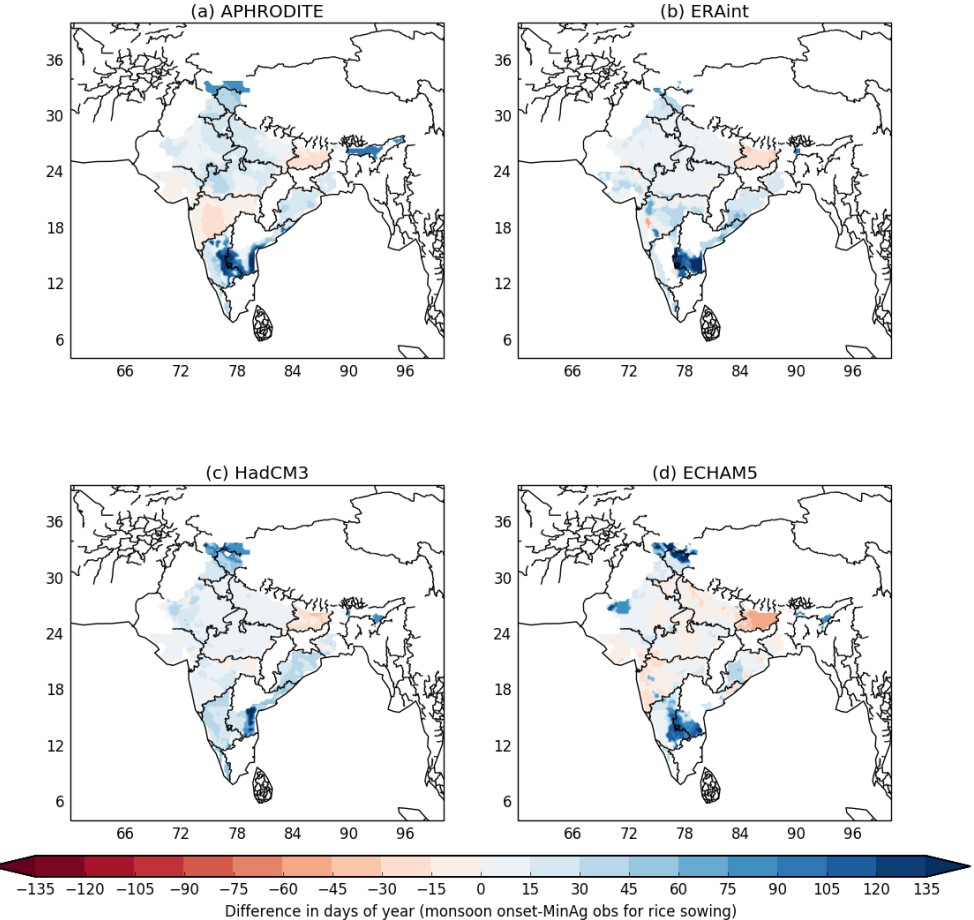

**Figure 5.** Plots of the difference between the midpoint of the monsoon onset in the model and the midpoint of the observed rice sowing period for 1990-2007.

The monsoon onset and retreat estimates are provided in days of year (pentads) therefore with a range of plus or minus 2.5 days. The MinAg observations are also provided in days of year with a range that varies from plus or minus 15 days depending on the location. Figure 8 shows the range of the MinAg sowing and harvest observations for each state; the full sowing or harvest window is shown by the downward grey triangles, with the midpoints shown by black triangles joined by a black line. Figure 6 considers the midpoints of these two ranges in order to summarize how well aligned the

monsoon onset range is to the observed range of rice sowing dates i.e. how the 5-day onset windows

coincide with the observed sowing window. If the monsoon onset range is completely within the range of sowing days provided by the observations then this is classed as a 'hit' (shown by the blue regions). If the monsoon onset range is completely outside the range of observed sowing days then this is classed as a 'miss' (shown by the red regions). The yellow regions in Fig. 6 show the places where the monsoon onset overlaps the range of observed sowing days but does not completely fall

within it; these regions are labelled 'Overlaps'. Figure 6 has only a small area of red indicating that monsoon onset is, for large parts of India, within the range of days of rice sowing. In each plot shown in Fig. 6 the region that is red or yellow is different, this makes it difficult to say if one dataset is better than another. ECHAM5 appears to have the smallest total area or red/yellow which is probably because ECHAM5 tends to have an earlier onset than the other datasets and in general that makes it

closer to the rice sowing dates. Table 1 lists the differences between the monsoon statistics (onset and retreat) and the relevant sowing and harvest dates for each crop calculated for each of the simulations and the APHRODITE observations and averaged for India. Table 1 shows the that on average across India rice sowing occurs between 10 and 20-days prior to the averaged modelled monsoon onset (3rd block, Table 1). We would not expect the different datasets to give the same results, however

Table 1 shows that they are relatively consistent with each other and importantly with observations as is illustrated by the APHRODITE data. Table 1 highlights that on average APHRODITE requires a larger croprule than the simulations for rice sowing, however this is not always the case for sowing or harvest and rice or wheat. The croprules used here are based on the 1990-2007 period for which ERAint has the earliest onset (see Fig. 10). ECHAM5 has the smallest croprule to move it towards

the rice sowing date but the highest variance in the mean difference between the monsoon onset and the MinAg rice sowing date. APHRODITE has the largest crop rule for rice sowing indicating that the weighted average of the APHRODITE monsoon onset is further from the rice sowing date than for other datasets.

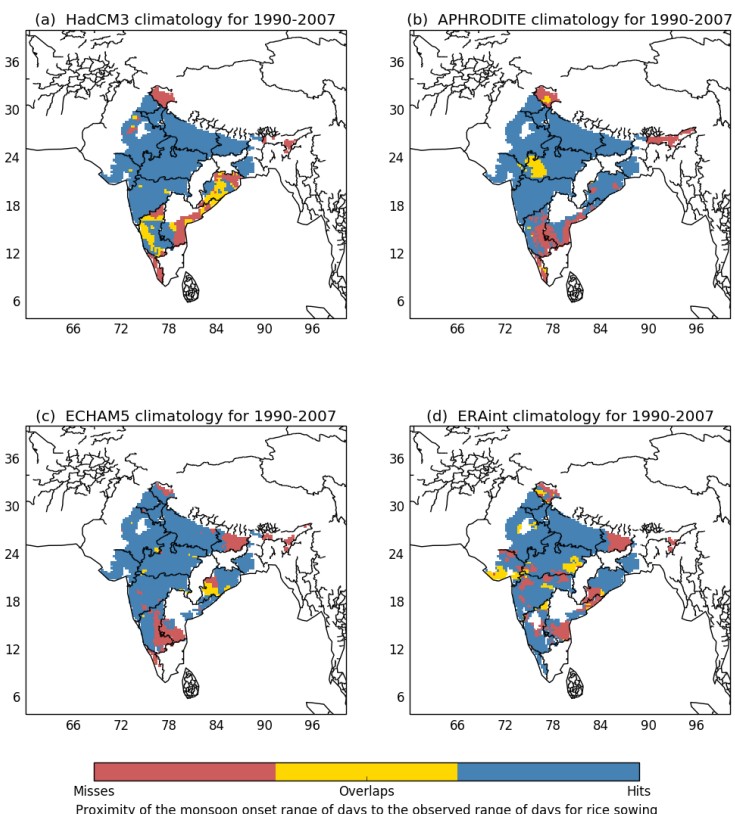

Plots of the comparison between model monsoon onset and observations for rice sowing

(a) HadCM3 climatology for 1990-2007    (b) APHRODITE climatology for 1990-2007

(c) ECHAM5 climatology for 1990-2007    (d) ERAint climatology for 1990-2007

Misses      Overlaps      Hits

Proximity of the monsoon onset range of days to the observed range of days for rice sowing

**Figure 6.** The comparison of the model monsoon onset in terms of the days of the year (to within the pentad) and the range of days of the year for the observed sowing date for rice. This is shown in terms of hit (blue) and overlap (yellow) or if there was no overlap this is shown as a miss (red)

In general the differences between rice harvest and monsoon retreat are larger but still consistent
across the region (see Fig. B.1), with rice harvest occurring on average 30-40 days after monsoon retreat (see 4th block, Table 1). Wheat sowing tends to occur approximately 60-70 days after monsoon retreat (see Fig. B.2 and 1st block, Table 1) and wheat harvest tends to occur approximately 90-101 days before monsoon onset (see Fig. B.3 and 2nd block Table 1). These values (given in Table 1) provide the $RelMonsoon_{croprule}$ values introduced in Sect. 2.3.1 used to adjust the monsoon
statistics and calculate the new sowing and harvest dates based on the monsoon. There are small regions with different monsoon characteristics and therefore much earlier sowing days, for example for rice sowing in the southern and far north of India. These regions have a direct impact on the values (minimum, maximum, mean and standard deviation - SD) given in Table 2 which are averages for the whole of India and are discussed in more detail in Sect. 4. Fig 1 highlights that the the average
sowing and harvest dates for rice and wheat are closely aligned with the monsoon precipitation from all three RCM simulations.

### 3.2 Monsoon derived estimates of sow/harvest dates for rice and wheat

The monsoon derived sowing and harvest dates are calculated from applying the $RelMonsoon_{croprule}$ for each model (See Table 1) to the simulated monsoon onset and retreat fields (see Fig.2). Here we compare these with the gridded observations to see how well the method performs for the present day. The monsoon derived sowing and harvest dates are compared with the MinAg observations using regional maps and an analysis for each state area in order to show the differences in the method across India.

Figure 7 shows the monsoon derived estimates of rice sowing dates (left column) and compared with MinAg observations (right column). Fig. C.1 shows the same plots for rice harvest, with plots for wheat shown in Fig.C.2 and Fig. C.3 for sowing and harvest respectively. The $RelMonsoon_{croprule}$ for wheat for both sowing and harvest are much larger than those for rice but there is still good agreement between the monsoon derived estimates and the MinAg observations across the region. On average the monsoon derived estimates of sowing and harvest dates are within 4 days of the midpoints for the sowing and harvest dates for rice and within 7 days of the midpoints for sowing and harvest dates for wheat. There is some variation across India with some regions showing some larger differences but generally the monsoon derived estimates for sowing and harvest dates are within the range provided by the observations across much of the region for both crops.

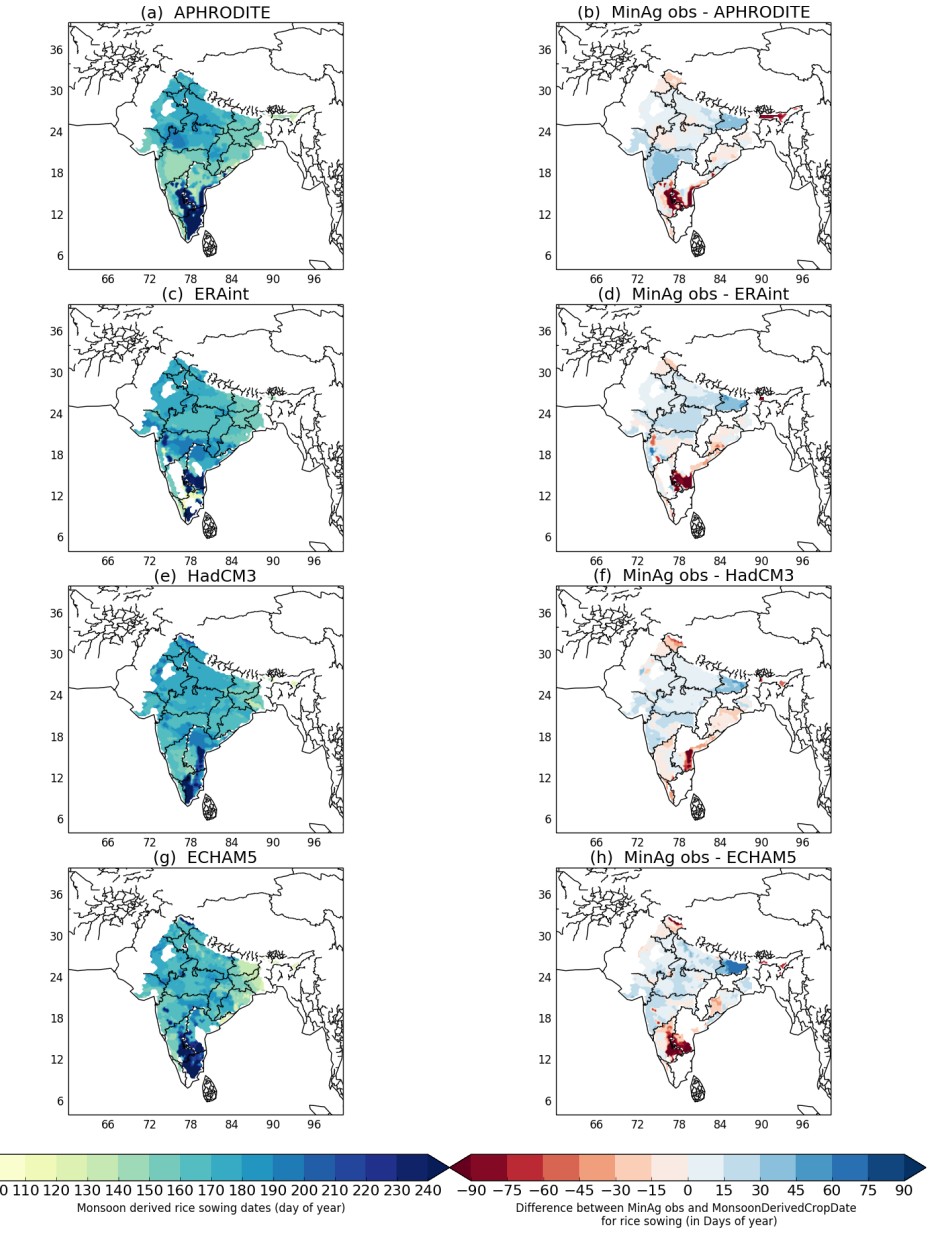

**Figure 7.** The monsoon derived rice sowing dates (left) and the difference between the MinAg observations and the monsoon derived rice sowing dates (right) for the period 1990-2007.

Figure 8 shows the average crop duration for each state where MinAg observations were available
for the 1990 to 2007 period alongside the crop duration for each of the four sets of monsoon derived
estimates using the Fig. 2 method. In the majority of states shown in Fig. 8 the sowing and harvest
dates calculated using the Fig. 2 method were within the range of the MinAg observations for rice
and wheat sowing and harvest dates, however the overall performance was better for rice compared
with wheat and sowing compared with harvest in each crop. Figure 8 also highlights the difference

in both the observed and simulated crop duration between the two crops with rice having a shorter season than wheat. In general across most of the states with available data the method provides a reasonable estimate of the sowing, harvest date and crop duration. Even where the method does not quite capture the observed sowing and harvest dates, the method is often just outside the observed range.

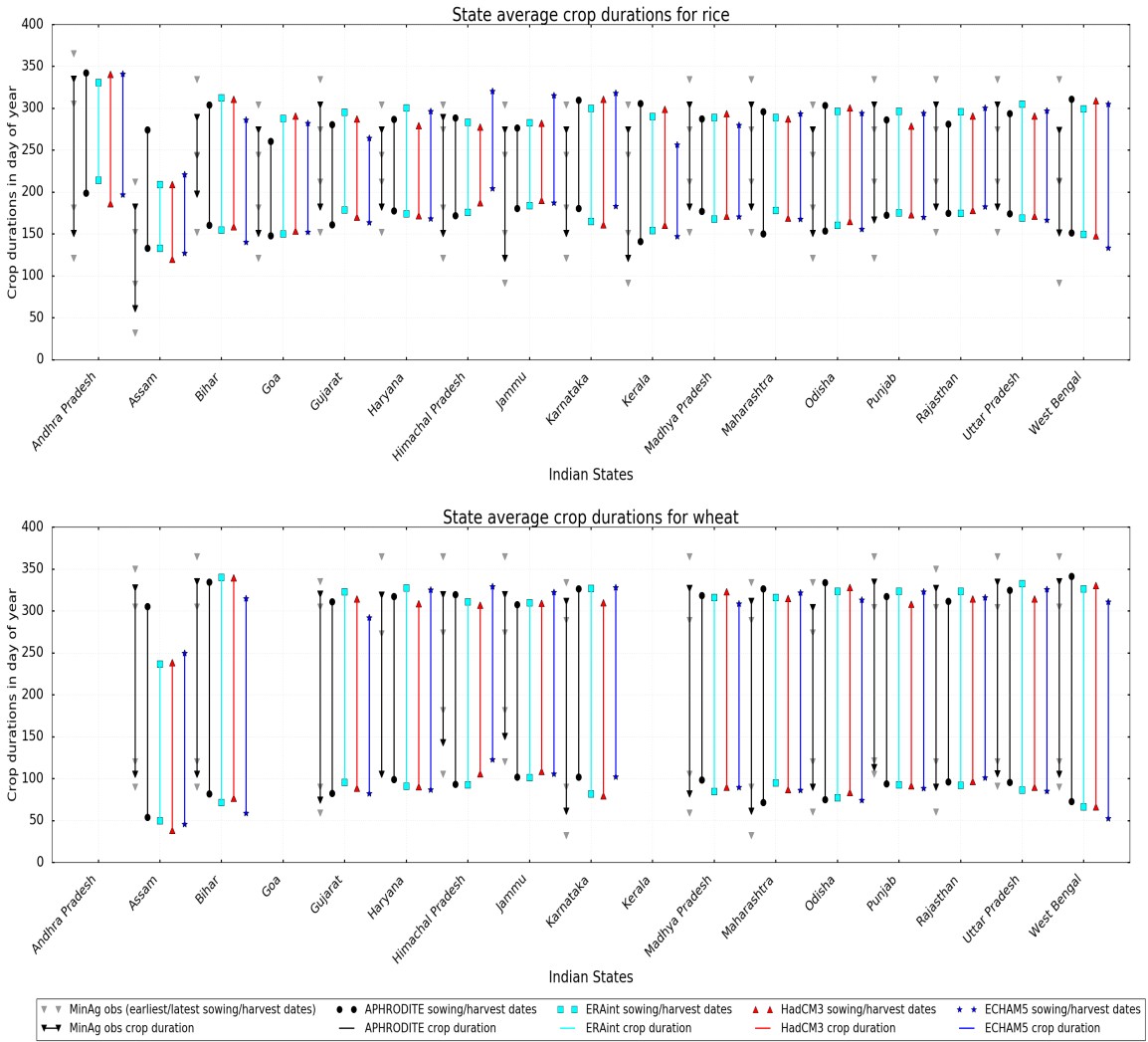

**Figure 8.** The state averaged crop durations for each dataset are shown by the lines for each state together with the sowing and harvest dates shown by the different shapes at the end of each line. The MinAg observations are shown by the black line and downward triangles, with the paler triangles representing the full range of sowing and harvest days for that state. The APHRODITE observations are also shown by black lines and filled circles for the sowing and harvest dates. ERAint is shown by cyan lines and squares, ECHAM5 by blue lines and asterisks, HadCM3 by red lines and upward triangles.

In order to establish how well the method performs over all, we use Fig. 8 to assess if the results using the method are good, poor or fair compared to the MinAg data. Where the monsoon derived sowing and harvest dates from three of the four datasets using the method are within the range of the MinAg data as shown in Fig. 8, the results of the method are said to be 'good' for a state. The results of the method are said to be 'fair' where two datasets are within the range of the MinAg data and 'poor' where the sowing and harvest dates fall outside the observed range. In this analysis only the state of Assam did not have any 'good' scores for rice or wheat sowing or harvest. Most of the scores for most of the states for both sowing and harvest, and wheat and rice had a score of good or fair.

In general the regions where the monsoon derived sowing and harvest dates are not as close to the MinAg observations tends to be for the states in the south, such as Andhra Pradesh and Karnataka or to the north of India, such as Jammu and Himachal Pradesh. This is supported by the maps, particularly for rice for these regions (in Fig. 7 and Fig. C.1) which show that the method does not perform as well for some of these states. These differences may be explained by the differing monsoon characteristics in these regions compared to the rest of India; these are highlighted in Fig. 3 and discussed further in Sect. 3.1 and Sect. 4. Assam in the north east of India is also noticeable compared with the other states in Fig. 8 with the rice crop season in the MinAg data displaced to an earlier part of the year. Assam tends to plant predominantly rice, tending to have three distinct rice seasons (autumn, winter and summer) rather than a rice-wheat rotation (Sharma and Sharma, 2015). In this analysis we use data for the Kharif paddy rice crop from the MinAg dataset which is planted and harvested earlier in Assam than in other states, with sowing in Feb/March and harvest in June/July (Bodh et al., 2015).

### 3.3 Analysis of future monsoon onset and retreat

As a demonstration of the method summarised in Fig. 2, the HELIX SWLs (described in Sec.2.4) are used to select two future periods: 2040-2057 and 2080-2097. Considering only these future periods, spatially HadCM3 and ECHAM5 show quite different future climates. HadCM3 shows a similar onset to the present day for 2040-2057 (see Fig. 9 (a) and (c)) but later onset compared with the present day for 2080-2097 (see Fig. D.1 (a) and (c)). ECHAM5 shows an earlier onset compared with the present day for the 2040-2057 period (see Fig. 9 (b) and (d)) but much later for the 2080-2097 period (see Fig. D.1 (b) and (d)). This suggests high variability in monsoon onset in these simulations. In fact all of monsoon onset, peak, retreat and duration show a large degree of variability as shown in Fig. 10 where each statistic has been averaged for South Asia. Each point in Fig. 10 represents a 17-year timeslice from between 1970 and 2097 for each of the APHRODITE, ECHAM5, HadCM3 and ERAint datasets. Figure 10 supports the points made regarding the spatial plots and also shows how the four monsoon statistics change between the 17 year timeslices. The 2040-2057 period has a much earlier onset for ECHAM5 than all the other periods except the 2000-

2017 period, which is similar (See Fig. 10 (a)). For most of the periods ECHAM5 has an earlier onset than HadCM3, this is also true of the retreat (See Fig. 10 (b)), the duration is usually longer for ECHAM5 compared with HadCM3 (See Fig. 10 (d)).

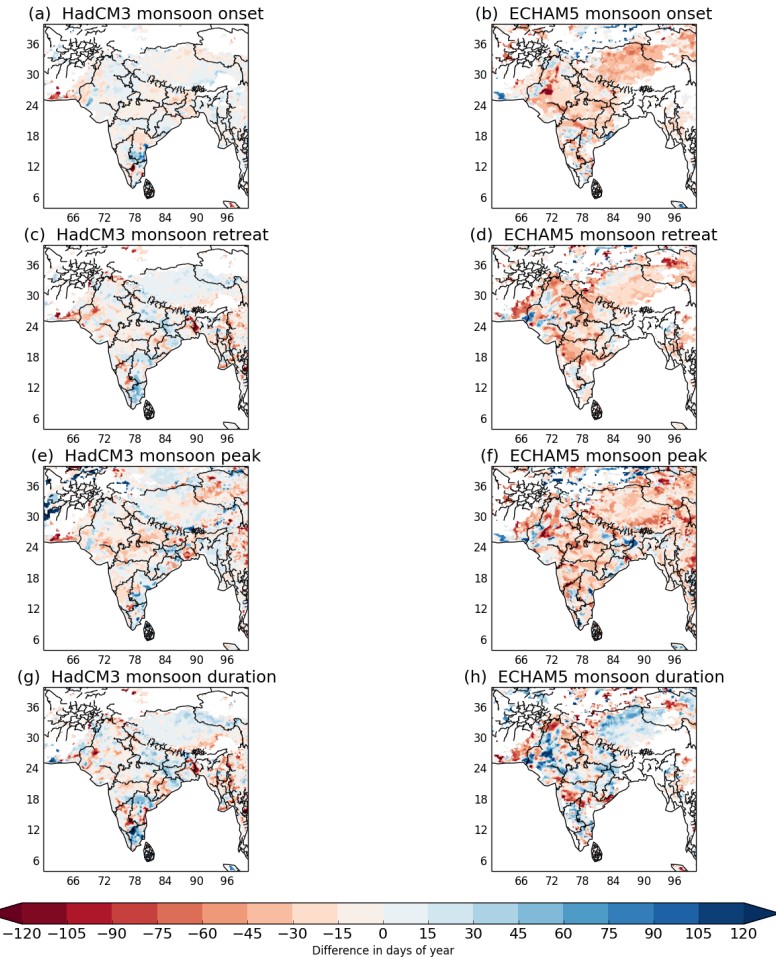

**Figure 9.** The difference between the monsoon statistics for the 2040-2057 future period and the present day 1990-2007 for HadCM3 (left) and ECHAM5 (right).

In order to illustrate the method for deriving sowing and harvest dates, Fig. 11 shows the annual
cycle of precipitation averaged for South Asia for the two future periods (plot a shows 2040-2057 and plot b shows 2080-2097) in the same way as the present day is shown in Fig. 1. The crop sowing and harvest dates used to provide the growing season durations in each of the plots shown in Fig. 11 for each of the simulations are calculated using the method described in Fig. 2. This shows that the proposed method provides an estimate of sowing and harvest dates that ensures the crops can
continue to be grown, in the simulation, when the climate is most appropriate rather than being fixed to the present day observed values.

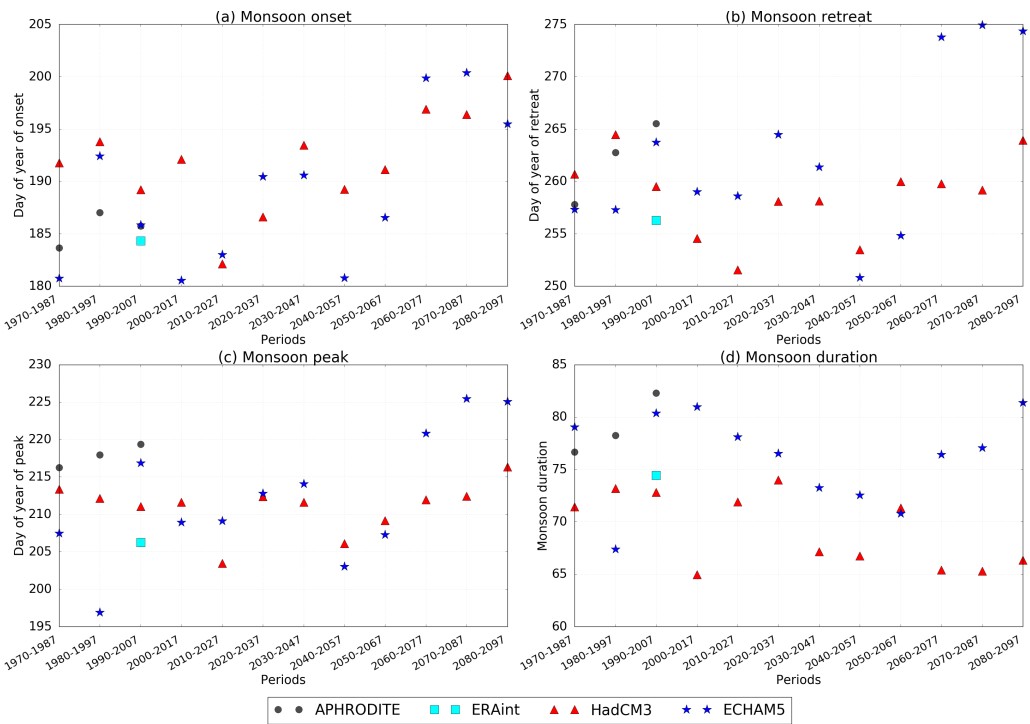

**Figure 10.** Monsoon statistics; onset (a), retreat (b), peak (c) and duration (d) averaged for South Asia for twelve 17-year timeslices between 1970-2097 to provide a timeseries of values for the region to assess the variability of the monsoon

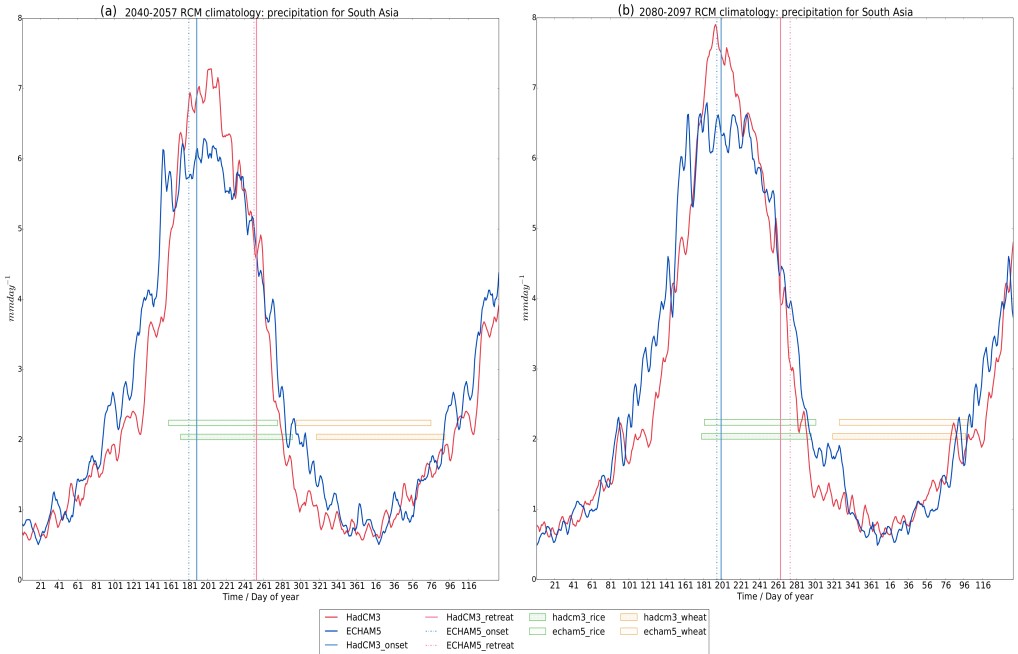

**Figure 11.** The one and a half year precipitation climatology for the period 2040-2057 (a) and the 2080-2097 (b) averaged for the whole of South Asia for each simulation (HadCM3-red line, ECHAM5-blue line) using a 5-day smoothed rolling mean. Also shown are the monsoon derived growing seasons for wheat (orange) and rice (green) calculated using the method described in Fig. 2 for HadCM3 (perpendicular hatching) and ECHAM5 (diagonal hatching). The monsoon onsets for each simulation are shown using blue vertical lines and retreat pink vertical lines (ECHAM5-dash dot lines, HadCM3-solid

## 4    Discussion

Recent climate impact studies such as AgMIP (Rosenzweig et al., 2013, 2014)) and ISIMIP (Warsza-wski et al., 2013, 2014) have highlighted the importance of reliable input data for models. Section 1.1 highlights the scale of the uncertainties present when solely using a global sowing and harvest dataset to simulate region specific cropping patterns. We have therefore proposed a new method for generating sowing and harvest dates for South Asia based on the ASM. The method reproduces observed sowing and harvest dates for much of India, these results are discussed further in Sect. 4.1. This method will also be useful in other monsoon regions where data are scarce, unreliable or unavailable such as in future climate simulations. The future results are discussed further in Sect 4.2.

### 4.1    Present day analysis

In general the method described by Fig. 2 works well across most of India for the present day, with the monsoon derived estimates of sowing and harvest dates falling within the range of days for sowing given by the observations and therefore providing a good estimate of the crop duration for most states (see Fig. 8). However there are regions where the estimated sowing and harvest dates do not compare as well against present day observations. Rice sowing is generally closely associated with ASM onset across most of central India, however in the south of India there is a small region where the differences between the observations of sowing dates and the monsoon are larger than everywhere else (see Fig. 5). In Sect. 3.1 this region is shown to have different monsoon characteristics to the rest of India. This part of India includes the state of Tamil Nadu, this state is located on the lee side of the Western Ghats and therefore does not receive the large amounts of ASM rainfall that is more commonly associated with this part of the world. Tamil Nadu receives up to 50 percent of its annual rainfall during October-December via the less stable North Eastern (NE) Monsoon. The NE monsoon is therefore more important for water resources for this part of India than the ASM which accounts for approximately 30 percent of the annual rainfall for this region (Dhar et al., 1982). These differing monsoon characteristics mean different agricultural practices are required to cultivate rice in this part of the country. This is illustrated by Fig. 12 (left plot) which shows that the southern region of India with differing monsoon characteristics irrigates rice more intensively than other parts of India. In the Tamil Nadu region, rivers are usually dry except during the monsoon months and the flat gradients mean there are few locations for building reservoirs, therefore approximately one third of the paddy rice crop is irrigated from a large network of water tanks (Anbumozhi et al., 2001). The Southern states of India have the highest density of irrigation tanks with large numbers also found in Andhra Pradesh and Karnataka, these are also regions shown to have a high irrigation intensity in Fig. 12. Rice harvest is typically not as closely associated with the monsoon onset as rice sowing, which usually requires the monsoon to be fully established before planting.

The widespread irrigation of wheat shown in Fig. 12 (right plot) has less of an impact on the estimates of wheat sowing/harvest dates because this crop is less closely linked to the monsoon onset than rice. Therefore the regional differences between the MinAg observations and the monsoon
derived sowing and harvest dates for wheat are not as large as some of those for rice (see Sect. 3.2). Given that the method has provided reasonable estimates of sowing and harvest dates for most of India, it would be useful and interesting to extend this method to improve it for the South of India.

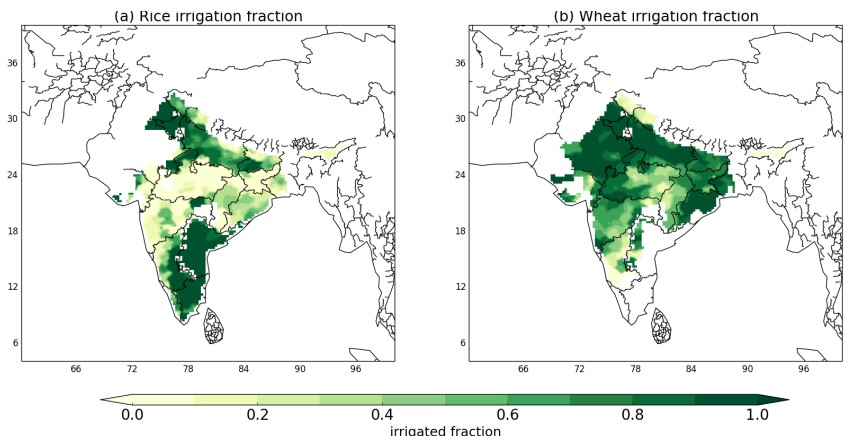

**Figure 12.** The average irrigation fraction for rice (a) and wheat (b) calculated from the ICRISAT observations of irrigation area and area planted

## 4.2 Future analysis

Analysis of the future monsoon onset, retreat, peak and duration shown in Sect. 3.3 shows how
changeable the ASM is for these simulations between time periods. Christensen et al. (2013) shows that there is a high model agreement within the ensemble from the 5th Coupled Model Intercomparison Project (CMIP5) for an earlier onset and later withdrawal in the future and therefore indicates a lengthening monsoon duration. However the simulations presented here do not show this with Fig. 10, instead highlighting the large amount of variability in the ASM for this region. It is possible that
an increase in the monsoon duration does occur in these simulations for some parts of South Asia but this detail is lost through averaging over the region or as a result of the time periods selected. Christensen et al. (2013) also suggest that there is medium confidence within the CMIP5 ensemble that the ASM rainfall will increase to the end of the century. The simulations presented do indicate this as shown by the timeseries in Fig. 13.

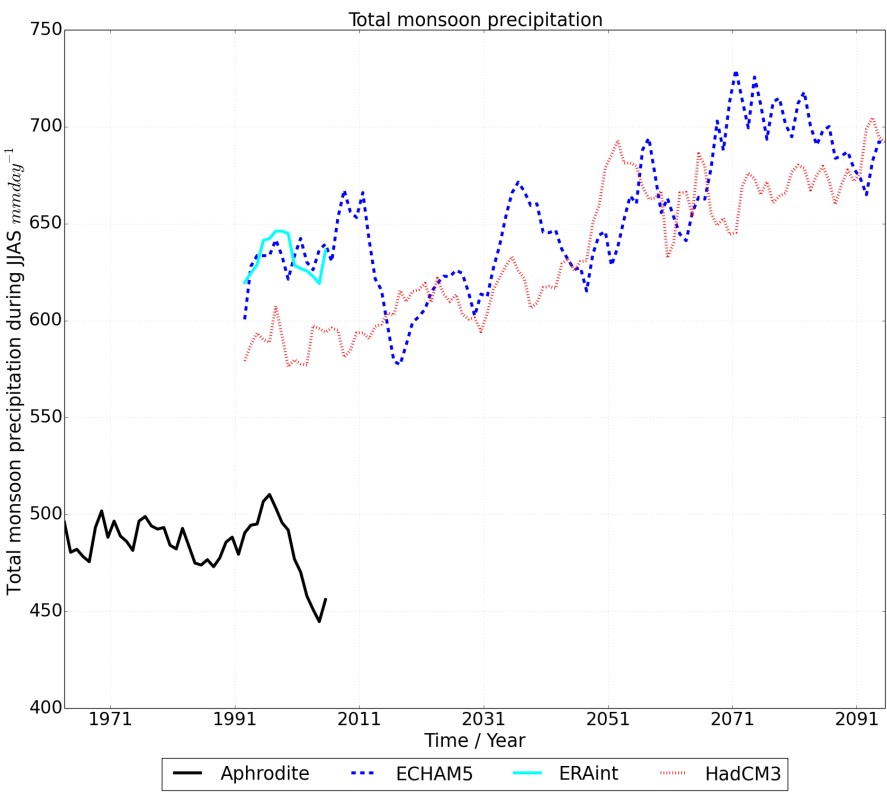

**Figure 13.** The annual timeseries of total monsoon precipitation, smoothed - using 5yr averaging, averaged for the whole of South Asia for all simulations; APHRODITE-solid black line, ERAint-solid cyan line, ECHAM5-blue dashed line and HadCM3-red dotted line.

Assuming that crops continue to be grown in accordance with the monsoon, Sect. 3.3 shows that the method described in Sect. 2 provides a good estimate of sowing and harvest dates for the two future periods shown. Spatial plots of the sowing and harvest dates for the two future periods (not shown) are similar to those in Sect. 3.2 for the present day with the south of the Indian peninsula continuing to show different monsoon characteristics (see Sect. 4.1) to the rest of India in the future, resulting in later estimated sowing and harvest dates for this region.

The proposed method successfully adjusts the sowing and harvest dates where the monsoon begins earlier in the future simulations and therefore provides a good estimate of sowing and harvest dates for the two future periods considered. This is a key benefit of using this method as it simulates the decision a farmer might take to sow before the usual observed date if the monsoon arrived early. This method therefore provides the capability for climate simulations to replicate the type of adaptation response that would happen in the real world. This method would also be useful for other regions that have a crop calendar that is similarly defined such as the SSA; this is a multiple cropping region with sowing and harvest dates closely associated with the main rainy season (Waha et al., 2013).

## 5 Conclusions

Sowing and harvest dates are an important input within crop models but are a source of considerable uncertainty. Global datasets, such as Sacks et al. (2010), cannot always distinguish when wheat is grown in tropical and sub-tropical regions therefore driving a requirement for higher resolution regional datasets. Crops across much of South Asia are heavily dependent on the ASM and therefore sowing and harvest dates tend to be closely linked to this climatological phenomena. We have therefore presented a new method for deriving sowing and harvest dates for rice and wheat for South Asia from the ASM onset and retreat. For the present day, the method generally shows good results for most areas of India with the derived sowing and harvest dates within the range of the observations for most states. The method does not work as well for the south of the Indian peninsular, this region receives a lower proportion of annual rainfall from the ASM than much of the rest of South Asia and irrigates intensively. Monsoon derived estimates of sowing and harvest dates for rice and wheat are useful for regions where data are scarce, unreliable or in future climate impact assessments. The method presented assumes that the agricultural practices remain dependent on the monsoon in the future. Given this assumption, the method presented successfully estimates the sowing and harvest dates for two future periods by adjusting the sowing and harvest dates according to the timing of the monsoon. Future work in this area could investigate refinements to the method to take into account the different characteristics of the monsoon in the regions where the method does not work as well and the differing agricultural practices there. It would also be interesting to investigate how well the method works for different crop rotations in different monsoon regions.

## Appendix A: Details of the models used

This analysis uses two General Circulation Models (GCMs) selected to capture a range of temperatures and variability in precipitation similar to the AR4 ensemble for Asia (Christensen et al., 2007) and the main features of the ASM (Kumar et al., 2013; Annamalai et al., 2007; Mathison et al., 2013, 2015). HadCM3; the Third version of the Met Office Hadley Centre Climate Model (HadCM3 – Pope et al., 2000; Gordon et al., 2000, a version of the Met Office Unified Model) provides the positive variation in precipitation and ECHAM5, (Roeckner et al., 2003, 3rd realization–) the negative variation in order to estimate the uncertainty in the sign of the projected change in precipitation over the coming century.

One RCM, the HadRM3 RCM (Jones et al., 2004) is used to downscale the GCM data to provide more regional detail to the global datasets. HadRM3 has 19 atmospheric levels and the lateral atmospheric boundary conditions are updated 3 hourly and interpolated to a 150 s timestep. These simulations include a detailed representation of the land surface in the form of version 2.2 of the Met Office Surface Exchange Scheme which includes a full physical energy-balance snow model (MOSESv2.2, Essery et al., 2003). MOSESv2.2 treats subgrid land-cover heterogeneity explicitly

with separate surface temperatures, radiative fluxes (long wave and shortwave), heat fluxes (sensible, latent and ground), canopy moisture contents, snow masses and snowmelt rates computed for each surface type in a grid box (Essery et al., 2001). However the air temperature, humidity and wind speed above the surface are treated as homogenous across the gridbox and precipitation is applied uniformly over the different surface types of each gridbox (Mathison et al., 2015). This RCM was included in an assessment of four RCMs conducted by Lucas-Picher et al. (2011) for the South Asia region which demonstrated that RCMs were able to capture the monsoon.

HadRM3 is driven by boundary data from the two GCMs (See Fig.2) to provide 25 km resolution regional climate modelling of the Indian sub-continent (25° N, 79° E–32° N, 88° E) for the period 1960–2100. These RCM simulations are from the EU-HighNoon project (referred to hereafter as HNRCMs), representing currently the finest resolution climate modelling available for this region (Mathison et al., 2013; Moors et al., 2011; Kumar et al., 2013).

The HNRCMs use the SRES A1B scenario which represents a future world of very rapid economic growth, global population that peaks in mid-century and declines thereafter, and rapid introduction of new and more efficient technologies. The A1B scenario specifically, represents this future world where there is balance across energy sources i.e. a mixture of fossil and non-fossil fuels (Nakicenovic et al., 2000).

**Appendix B:  Comparing observed sowing and harvest dates with estimates of monsoon onset and retreat**

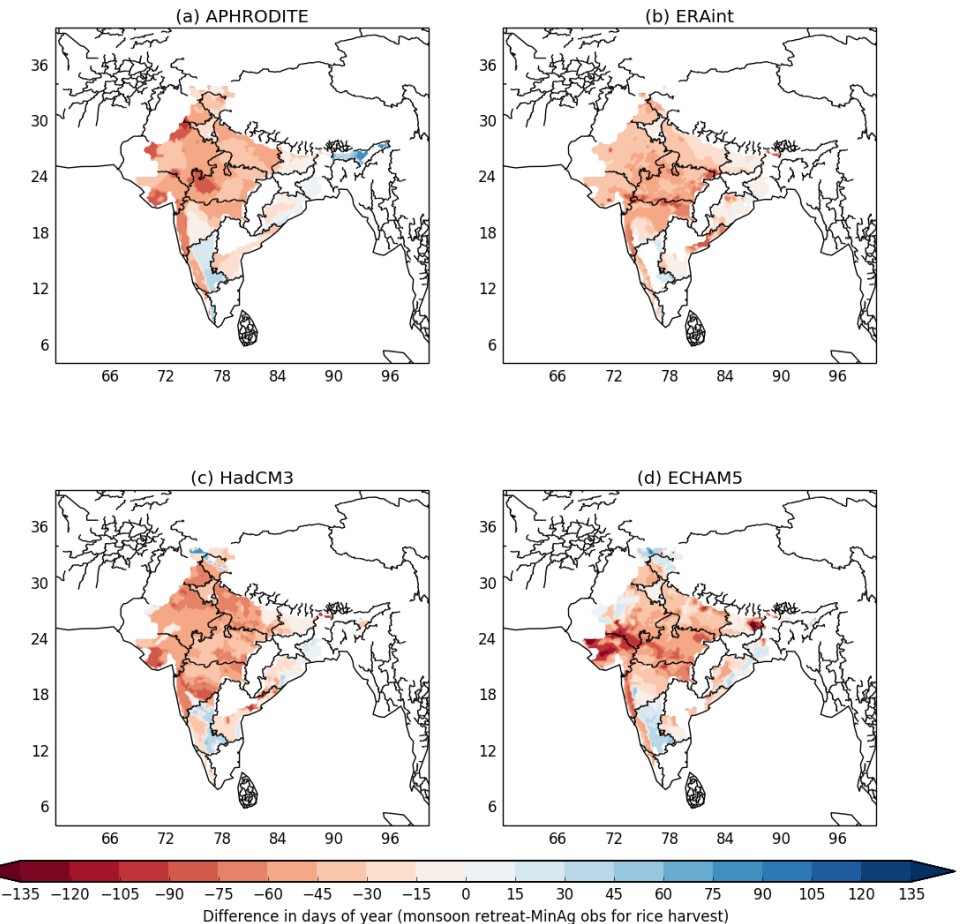

**Figure B.1.** The difference between the midpoint of the monsoon retreat in the model and the midpoint of the observed rice harvest period for 1990-2007.

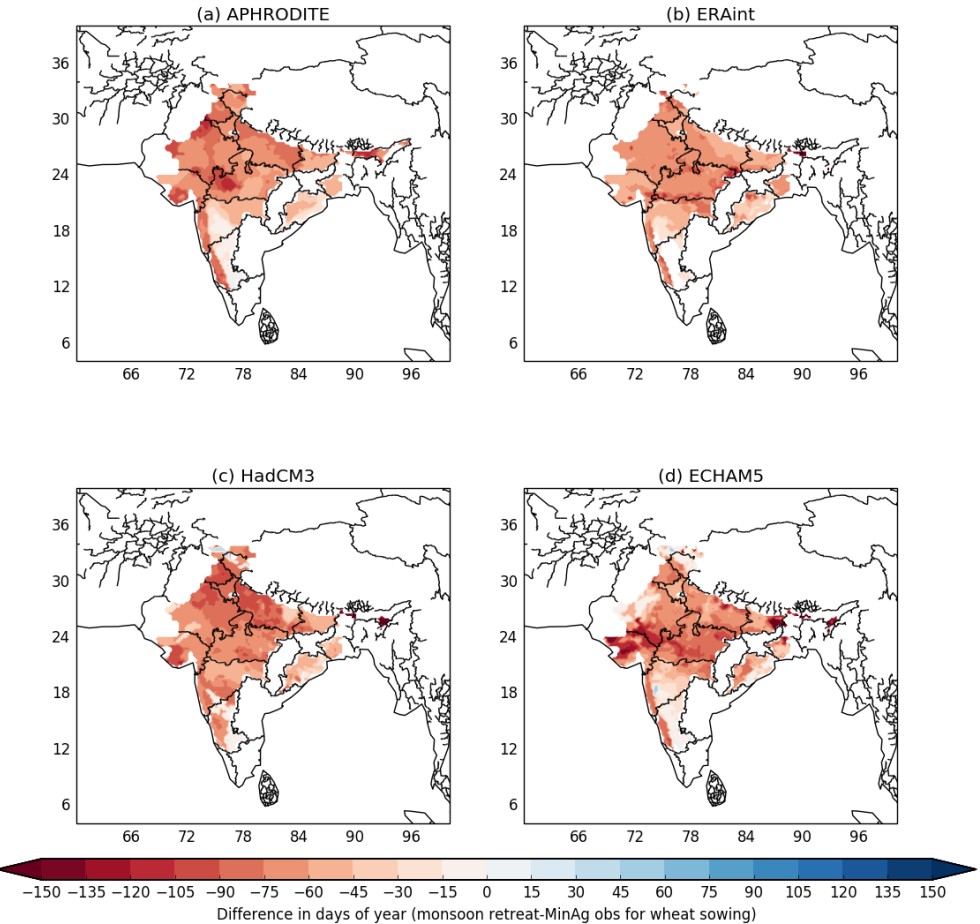

**Figure B.2.** The difference between the midpoint of the monsoon retreat in the model and the midpoint of the observed wheat sowing period for 1990-2007.

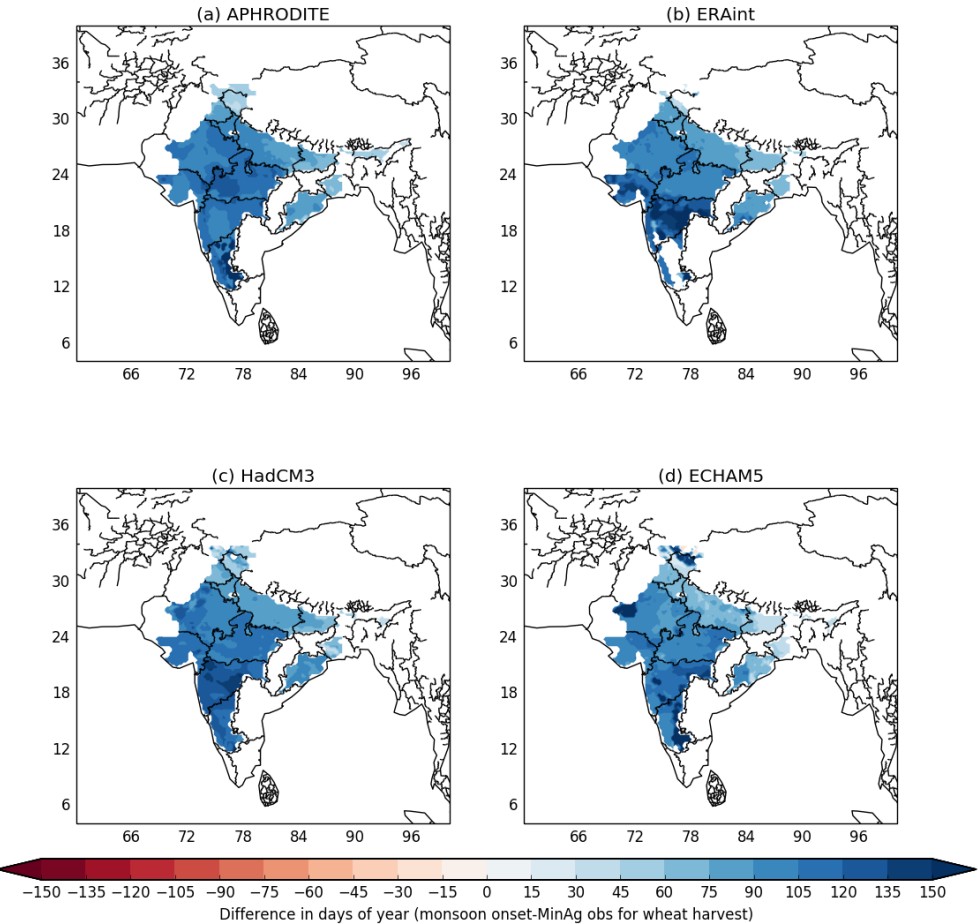

**Figure B.3.** The difference between the midpoint of the monsoon onset in the model and the midpoint of the observed wheat harvest period for 1990-2007.

**Appendix C: Monsoon derived estimates of sow/harvest dates for rice and wheat**

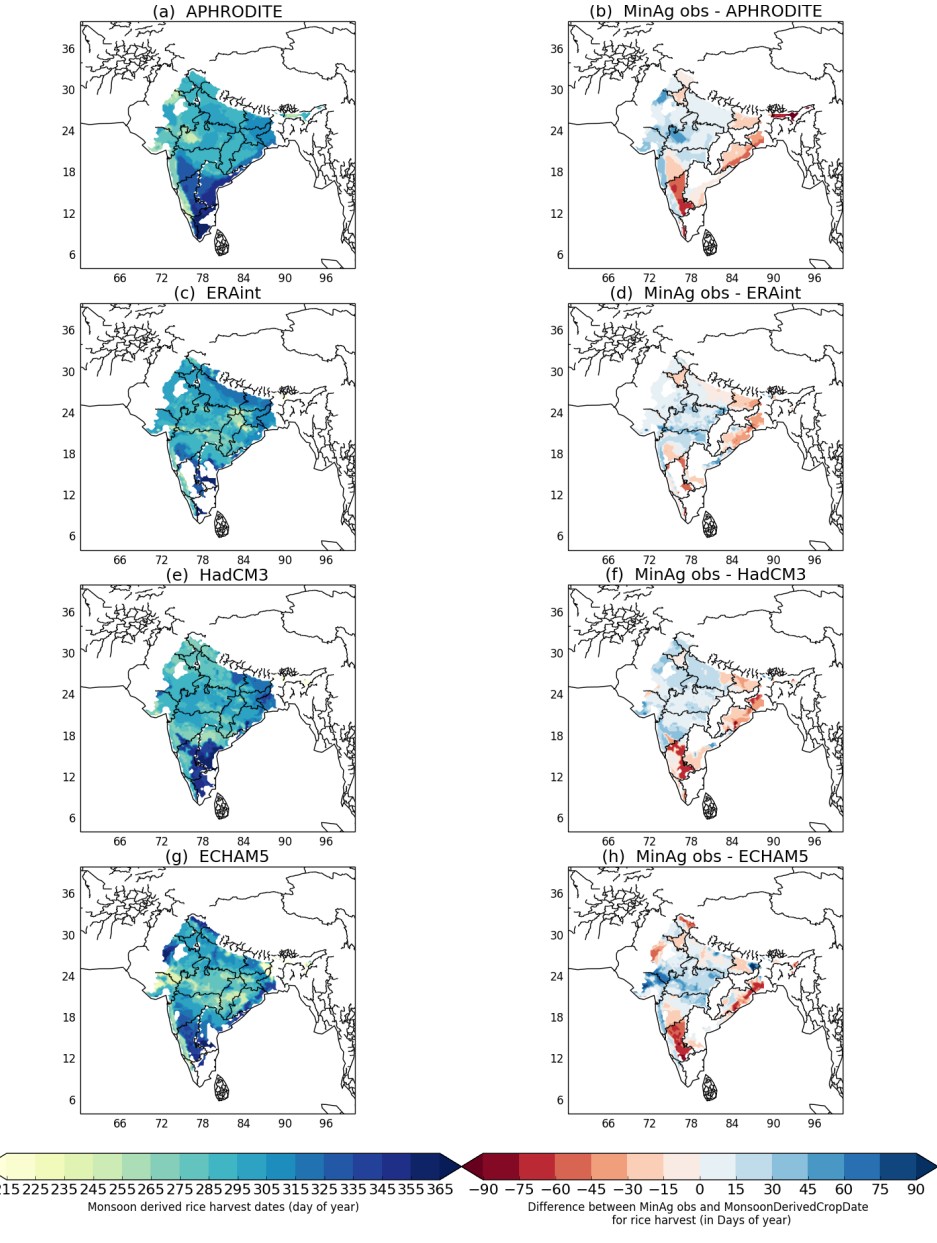

**Figure C.1.** The monsoon derived rice harvest dates (left) and the difference between the MinAg observations and the monsoon derived rice harvest dates (right) for the period 1990-2007.

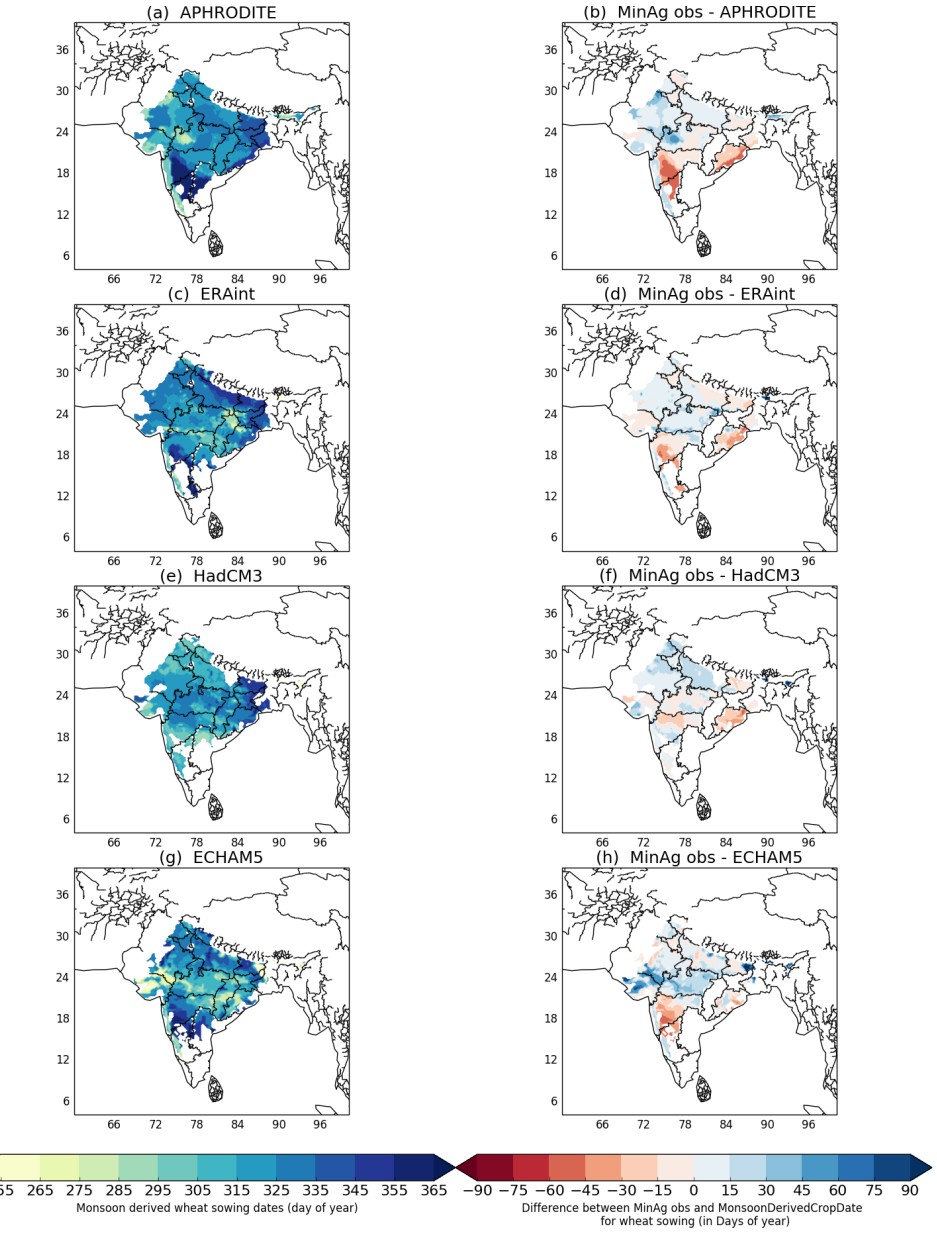

**Figure C.2.** The monsoon derived wheat sowing dates (left) and the difference between the MinAg observations and the monsoon derived wheat sowing dates (right) for the period 1990-2007.

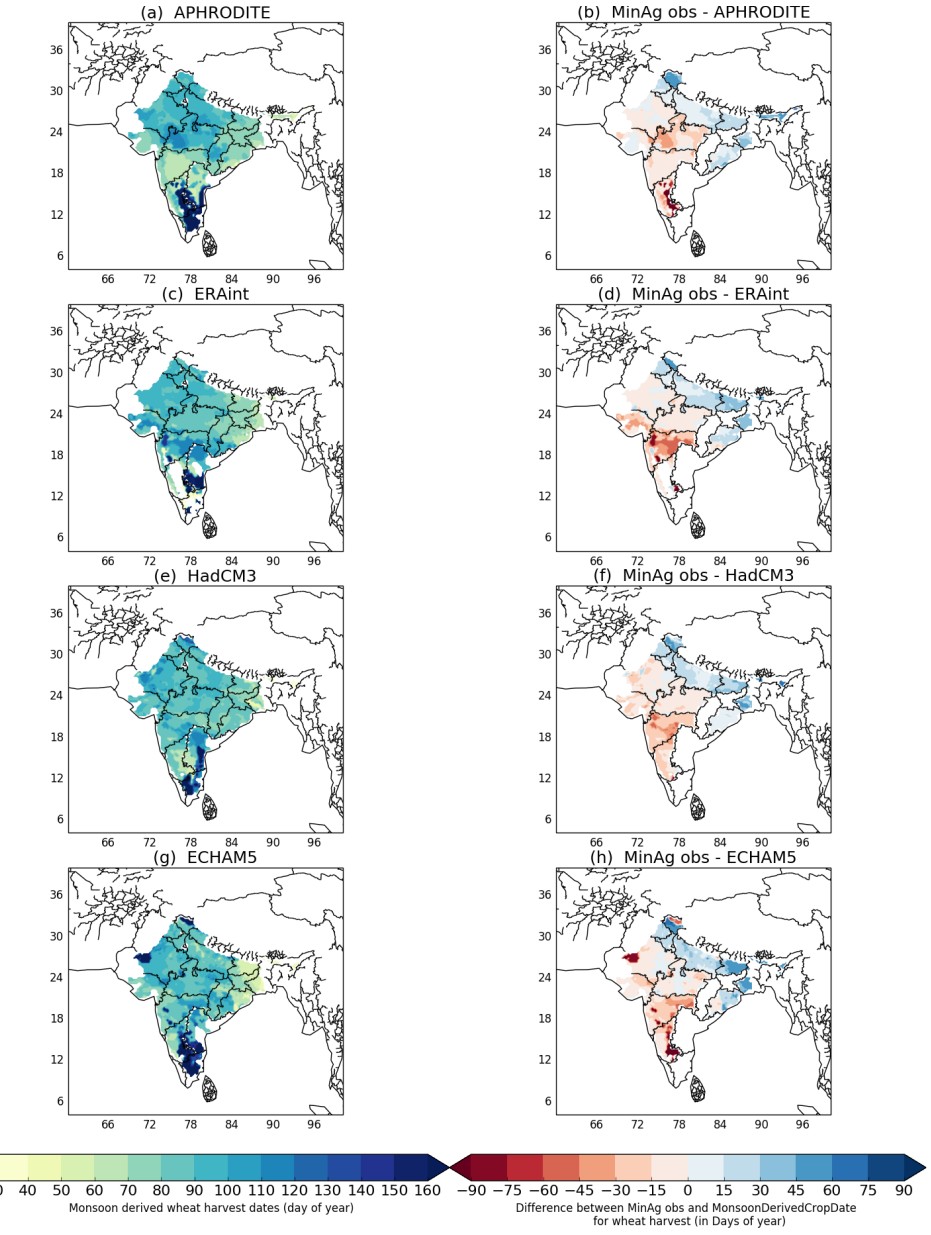

**Figure C.3.** The monsoon derived wheat harvest dates (left) and the difference between the MinAg observations and the monsoon derived wheat harvest dates (right) for the period 1990-2007.

## Appendix D: Analysis of future monsoon onset and retreat

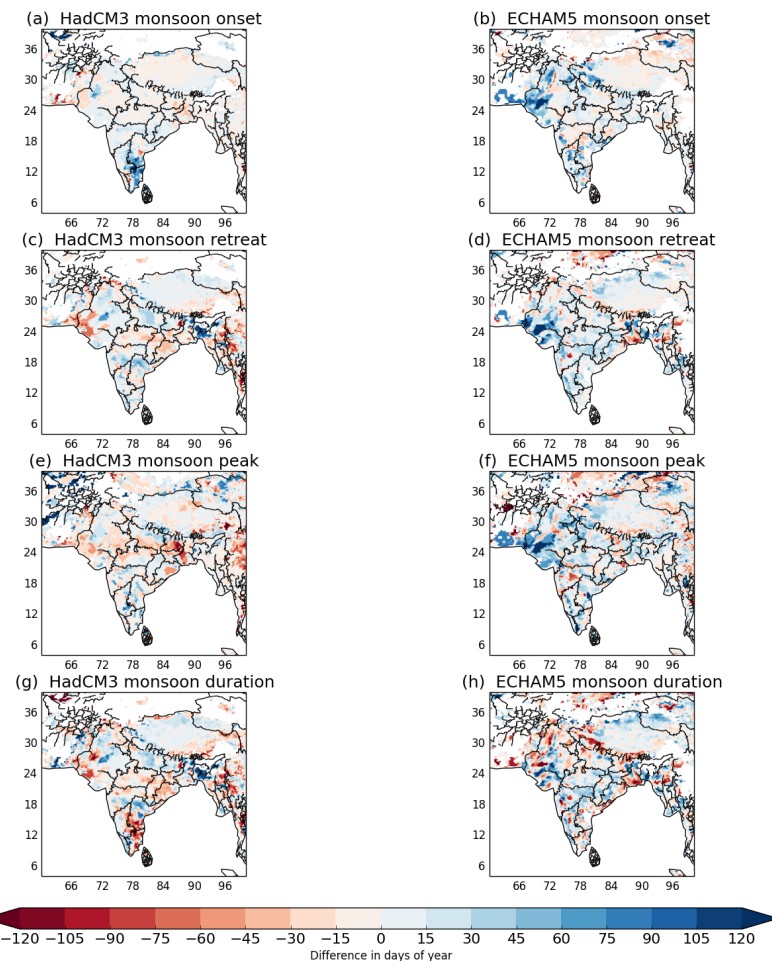

**Figure D.1.** The difference between the monsoon statistics for the 2080-2097 future period compared with the present day 1990-2007 for HadCM3 (left) and ECHAM5 (right).

*Acknowledgements.* The research leading to these results has received funding from the European Union Seventh Framework Programme FP7/2007-2013 under grant agreement no. 603864. Camilla Mathison was supported by the Joint UK DECC/Defra Met Office Hadley Centre Climate Programme (GA01101). Thanks to Andy Wiltshire for the initial discussions that contributed to the original idea, Gill Martin for reviewing code and helping with the development of the existing monsoon statistics code into Python.Thanks also to Karina
Williams for some valuable discussions, help with Python code and review comments.

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

**Table 1.** Table of $RelMonsoon_{croprule}$ for each dataset, crop and stage. The $RelMonsoon_{croprule}$ is the value subtracted from the monsoon onset/retreat in order to calculate a new sowing/harvest date based on the monsoon onset/retreat. In each case the new estimate of the sowing and harvest dates is calculated by subtracting the $RelMonsoon_{croprule}$ from the $Mon_{stat}$ where $Mon_{stat}$ is Monsoon onset or Monsoon retreat from a HNRCM or APHRODITE precipitation observations. Where the sowing/harvest is **before** the monsoon statistic, the crop rule is in **bold** with normal type indicating that sowing/harvest occurs after the monsoon statistic.

| crop | stage | $Mon_{stat}$ | source | $RelMonsoon_{croprule}$ (India average) |
|------|-------|--------------|--------|------------------------------------------|
| wheat | sowing | retreat | APHRODITE | -63.5 |
| wheat | sowing | retreat | ERAint | -62.8 |
| wheat | sowing | retreat | HadCM3 | -67.9 |
| wheat | sowing | retreat | ECHAM5 | -63.6 |
| wheat | harvest | onset | APHRODITE | **98.5** |
| wheat | harvest | onset | ERAint | **100.4** |
| wheat | harvest | onset | HadCM3 | **98.9** |
| wheat | harvest | onset | ECHAM5 | **91.4** |
| rice | sowing | onset | APHRODITE | **19.7** |
| rice | sowing | onset | ERAint | **17.3** |
| rice | sowing | onset | HadCM3 | **17.2** |
| rice | sowing | onset | ECHAM5 | **10.1** |
| rice | harvest | retreat | APHRODITE | -32.7 |
| rice | harvest | retreat | ERAint | -35.4 |
| rice | harvest | retreat | HadCM3 | -38.5 |
| rice | harvest | retreat | ECHAM5 | -34.7 |

**Table 2.** Analysis of the differences between the midpoints of the MinAg data and Monsoon onset/retreat for rice/wheat sowing and harvest dates: The table shows the minimum, maximum, mean and standard deviation (SD) averaged across South Asia where wheat or rice are planted.

| crop | stage | monsoon stat | source | min | max | mean | SD |
|------|-------|--------------|--------|-----|-----|------|-----|
| wheat | sowing | retreat | APHRODITE | -122.0 | 53.0 | -63.5 | 23.6 |
| wheat | sowing | retreat | ERAint | -160.0 | 36.0 | -62.8 | 19.8 |
| wheat | sowing | retreat | HadCM3 | -185.0 | 33.0 | -67.9 | 26.7 |
| wheat | sowing | retreat | ECHAM5 | -187.5 | 53.0 | -63.6 | 34.6 |
| wheat | harvest | onset | APHRODITE | 32.5 | 216.5 | 98.5 | 26.5 |
| wheat | harvest | onset | ERAint | 22.0 | 216.5 | 100.4 | 26.8 |
| wheat | harvest | onset | HadCM3 | -3.0 | 216.5 | 98.9 | 23.0 |
| wheat | harvest | onset | ECHAM5 | -18.0 | 217.5 | 91.4 | 33.7 |
| rice | sowing | onset | APHRODITE | -24.5 | 156.5 | 19.7 | 32.8 |
| rice | sowing | onset | ERAint | -49.5 | 196.5 | 17.3 | 30.5 |
| rice | sowing | onset | HadCM3 | -40.0 | 226.5 | 17.2 | 25.4 |
| rice | sowing | onset | ECHAM5 | -65.0 | 186.5 | 10.1 | 36.7 |
| rice | harvest | retreat | APHRODITE | -91.5 | 110.5 | -32.7 | 30.4 |
| rice | harvest | retreat | ERAint | -116.5 | 73.5 | -35.4 | 23.3 |
| rice | harvest | retreat | HadCM3 | -111.5 | 78.5 | -38.5 | 29.3 |
| rice | harvest | retreat | ECHAM5 | -141.5 | 98.5 | -34.7 | 35.9 |