# Peer review of "Estimating sowing and harvest dates based on the Asian Summer Monsoon"

_Earth System Dynamics, 2017_

## Referee Comment (RC1) · Anonymous Referee #1 · 4 Dec 2017

**General Comments:**

This study attempts to establish crop sowing and harvest dates in India that are tailored to the seasonal monsoon – a regional climate system that dictates crop cycles for the majority of India's farmers. The authors motivate this study by 1) describing the several inconsistencies and limitations of larger-scale estimates of sowing/harvest dates and crop calendars, 2) summarizing the need to incorporate higher resolution products as we move towards more advanced, regional climate-crop assessments, and finally 3) detailing the sensitivities of Indian cropping systems – specifically rice-wheat systems – which requires rapid plot preparation and turnover between crops for the successful production of both crops. The authors also put forth an important point that it is unrealistic to hold crop planting/harvest constant in crop model simulations of future climate

impacts, as we expect farmers to adapt as weather and climate patterns change.

In general, this study is of good quality and is highly warranted (as the authors point out) to facilitate improved climate-crop model experimentation and to incorporate autonomous adaptation. The motivation and methods the authors provide are entirely reasonable. Methods like the ones presented here could be of great utility to the wider crop modeling community in that a) it could prove a more realistic representation of dryland farming systems and regions that are highly dependent on rainfall and b) the methods appear straightforward enough and flexible to facilitate wide implementation across models and similar monsoonal agricultural systems.

However, I do think certain clarifications in the methods are warranted to provide the reader with additional understanding as to why some metrics were selected and, importantly, to contextualize the results (e.g. In Table 1, what constitutes a reasonable amount of time between planting and harvest, as derived using model simulations and in light of the need to better represent these rice-wheat systems?). I think the authors might briefly include measures or notes of what constitutes a "reasonable" representation in the Methods section and apply these in the Results. There might be an added caveat as well on how these Methods really demand a "good" representation of the monsoon system (and what that means – with respect to Aphrodite or another product), and that therefore there are limitations in applying this to the wider range of climate models, which have historically had issues in simulating the monsoon components (including onset and intraseasonal variability).

As such, I recommend that this manuscript undergo revisions. I think if the authors can sufficiently address the following concerns, the manuscript would be of benefit and utility to the larger community.

Specific Comments:

Line 111: I appreciate the discussion of the limitations of the Sacks et al dataset to your applications, but is the Sacks dataset actually used in your Methods at all? It looks to
me that the Methods and Results are relative only to the Bodh (2015) dataset. If this is the case, then I think it can be cut from the figures and the section trimmed a bit (after it briefly stating the Sacks et al issues and that it's not appropriate for regional assessment needs).

Section 2.2: Developing metrics for defining site or sub-region specific onset/withdrawal periods can be a rather arbitrary procedure in defining "critical" thresholds. I think it's fine that the authors selected the NPPI – in looking at Figure 4 and roughly eye-balling IMD contours (http://www.imd.gov.in/pages/monsoon\_main.php?adta=JPG&adtb=1), there appears to be consistency. Figures 5 and 6 also show consistency between onset and sowing dates, which is good.

That said, I think a bit more discussion here on why this particular index was selected could be useful, particularly to people who work in the region and want more monsoon-tailored agro-climatic assessments. The original reference Zeng and Lu (2004), upon which the Lucas-Picher (2011) index is based, did indeed create the NPPI for a consistent way to evaluate important global monsoon statistics across datasets and models (despite the 0.618 threshold, which apparently did not have a physical basis).

However, given that this study is framed with respect to the importance of tailoring crop management information to the physical monsoon system, how does the NPPI provide that advantage to you over, say, another metric and/or threshold (Zeng et al. 2004; Li and Zhang 2009; Saini et al. 2011; Dong et al. 2015; Guimberteau et al. 2012; Zhang; Wang and Lin 2002; Goswami et al. 2006)? Were other indices considered and discarded? If so, why? The authors mention a few others in the beginning of the section but do not motivate the NPPI over the others, aside from saying what it is.

[Please note, the references provided on defining the onset are not comprehensive, and there is much more work here. Admittedly, some of the variables are data-limited or cannot be applied on the scales you seek. This is just to note there is a lot out there,
**so some added motivation would be good.]**

One way to better couch this section might be to simply move Section 3.1 to the Methods, rather than in the Results. Then you can say, for example, that you evaluate the NPPI across different models, and found the metric to be consistent with observed sub-regional monsoon onsets and with observed planting periods. You could then start the Results roughly from Section 3.3. Just a thought.

Lastly, Lucas-Picher (2011) note on their page 859 that the NPPI was used specifically as a standard with which to compare models, and that these periods may not specifically represent the beginning or end of the actual monsoon. Additionally and incidentally, they also note large range in NPPI across models, which led them to conclude that key physical monsoon processes may be missing from many model environments. How do the authors interpret these issues for their work here? Perhaps a statement should be added about the careful need to use models that are most representative of monsoon processes (not just that RCMs are better), and what the best measures are to evaluate this.

Section 2.3.1 Line 204: I may just be unable to find this information, but how was the "area averaging" applied to the RelMonsooncroprules? While the gridbox heterogeneity may be incorporated into the initial calculations of RelMonsooncroprule (as the authors mention in line 217), I wonder how the area-averaging actually impacts the authors' goal of capturing some of the spatial variation, which to me seemed a big goal of the study.

Section 2.3.1 Line 222-225: I would be careful to caveat this statement in relating it to the ASM, as the authors found that the methods needed to be adjusted for the northeast monsoon period in southern India (which will also affect Sri Lanka as well).

Figure 3: It would be helpful in Figure 3 to put "NPPI calculation" into box 2.3 and "RelMonsooncroprule"/MonsoonDerivedCropDate into box 2.4, to make those linkages explicit.

ESDD
Section 2.4 Line 229-231: This SWLs versus time period is a bit confusing given that regional temperature can exceed the SWLs, as noted. Perhaps it would be less confusing if the authors just picked one type of scenario, based either on time period or SWL.

Section 3.1 Line 261: How are you judging "compare well with observations"? Might a skill score or some other quantitative metric be included here? In fact, there are a couple of areas in the manuscript (e.g. Line 269 comparing the monsoon onset with the sowing date) where it is stated that the results look reasonable and compare well, but there is not much context given to what is meant by this.

I recommend moving Figure B1 to Figure 4, as it's a bit difficult to distinguish between the colors and having the differences with Aphrodite on hand would be more useful and quicker to interpret. Leave Aphrodite in the absolute colorbar however!

Figure 5: This is showing a difference in days, correct? So sowing in the lightest blue areas in the northwest of India are  $\sim$ 15 days ahead of the onset? If this is the case, then the colorbar title should be changed to "difference in days" or similar, not "Day of Year".

Section 3.2 Line 277: Table 1 seems to suggest that rice is sown 19 days before the onset as calculated using the Aphrodite dataset, if I'm reading that correctly (or at least there's an  $\sim$ 19 day gap between onset and sowing). How does this align with the comments in the introduction (Line 84), which said that "rice is usually sown with the first rains"?

I ask because this is also relevant to judging how "well" the onset and sowing periods overlap, and can be used to assess how well the method works. If the cropping systems are so tightly scheduled as they are described in the Introduction (Line 90-97), then this difference between sowing and onset could be impactful, no? What constitutes too much of a gap between sowing and onset?
And along these lines, I'm a bit curious about the ERAint and ECHAM5 results (Looking at Table 1 and Figure 6, for example). With respect to rice sowing, the India average results are much closer to the onset it seems (3.4 and 10.1, respectively). How do we qualitatively compare these results across models and observations?

In Figure 6 anyway, the coverage of blue makes ECHAM5 look - dare I say it – better than Aphrodite?! This is of course assuming that the Bodh et al dataset tells us something physical about the monsoon onset, in that farmers plant very close to or upon the onset (and thus sowing is a good proxy for the onset). A little guidance on how to interpret this would be appreciated, particularly with respect to the Aphrodite results (which I presume would be the most realistic?).

As an aside, it can be a bit confusing over the course of reading the paper to distinguish "before" or "after" relative to the monsoon statistics and sowing/harvesting. It's just a bit hard to keep the differencing straight. If there's a way to clearly and consistently indicate sowing dates that are before or after the onset, for example by scaling or color coding them explicitly, that would be very helpful.

Figure 6: This figure has potential to be very helpful, particularly related to my comments above, but is confusing in its current form and description. I think "hit", "overlap" and "miss" need to be more clearly defined, as there's an "overlap" color and the bar itself is called "overlap". The text, Lines 270-275, do not provide any further clarity in their current form, as both the word "around" and "at least close to" are used for the blue and the yellow. These are too vague to be helpful in interpreting the maps.

Technical Comments:

While I did not catch many technical faults with the manuscript, I would say in general, I the Intro and Motivation could be condensed a bit. It reads a bit long now and some of the material can be more succinctly stated.

Also "Section 2.3.1" may not be needed as you move directly to Section 2.4
The Discussion and Conclusions section can be condensed I think, as the beginning of the Discussion section (Lines 335-343) already read like the beginning to a conclusion.

I would lastly request that the authors check the titles of all their colorbars, as some (like Figure 5) I don't think guite convey the intent of the figure.

References:

Dong, G., H. Zhang, A. Moise, L. Hanson, P. Liang, and H. Ye, 2015: CMIP5 model-simulated onset, duration and intensity of the Asian summer monsoon in current and future climate. Clim. Dyn., 355-382, doi:10.1007/s00382-015-2588-z. Goswami, B. N., G. Wu, and T. Yasunari, The Annual Cycle, Intraseasonal Oscillations, and Roadblock to Seasonal Predictability of the Asian Summer Monsoon. http://journals.ametsoc.org/doi/pdf/10.1175/JCLI3901.1 (Accessed December 4, 2017). Guimberteau, M., K. Laval, A. Perrier, and J. Polcher, 2012: Global effect of irrigation and its impact on the onset of the Indian summer monsoon. Clim. Dyn., 39, 1329-1348, doi:10.1007/s00382-011-1252-5. Li, J., and L. Zhang, 2009: Wind onset and withdrawal of Asian summer monsoon and their simulated performance in AMIP models. Clim. Dyn., 32, 935–968, doi:10.1007/s00382-008-0465-8. Saini, R., M. Barlow, and A. Hoell, 2011: Dynamics and thermodynamics of the regional response to the Indian monsoon onset. J. Clim., 24, 5879–5886, doi:10.1175/2011JCLI3928.1. Wang, B., and H. Lin, 2002: Rainy Season of the Asian-Pacific Summer Mon-J. Clim., 15, 386–398. http://journals.ametsoc.org/doi/pdf/10.1175/1520soon. 0442%282002%29015%3C0386%3ARSOTAP%3E2.0.CO%3B2 (Accessed December 4, 2017). Zeng, X., E. Lu, X. Zeng, and E. Lu, 2004: Globally Unified Monsoon Onset and Retreat Indexes. J. Clim., 17, 2241-2248, doi:10.1175/1520-0442(2004)017<2241:GUMOAR>2.0.CO;2. http://journals.ametsoc.org/doi/abs/10.1175/1520-0442%282004%29017%3C2241%3AGUMOAR%3E2.0.00%3 (Accessed December 4, 2017). Zhang, H., Diagnosing Australia-Asian monsoon onset/retreat using large-scale wind and moisture indices. doi:10.1007/s00382-009https://link.springer.com/content/pdf/10.1007%2Fs00382-009-0620-x.pdf 0620-x.

ESDD
Printer-friendly version **Discussion** paper

Please also note the supplement to this comment: https://www.earth-syst-dynam-discuss.net/esd-2017-88/esd-2017-88-RC1supplement.pdf

---

## Referee Comment (RC2) · Anonymous Referee #2 · 12 Jan 2018

The authors have identified a relevant issue for improving broad scale climate impacts modelling for crops and adaptation studies. That being that using generic dates for crop sowing and harvesting are problematic, because sowing dates vary from year to year, and from location to location, and any simulation study that assumes they are uniform will be prone to potentially significant errors. They proposed a method to estimate sowing and harvest dates for two crops in sequence (in this case rice-wheat) using the onset of the Asian monsoon, and demonstrated performance of their new method. I think the topic is novel and important and is well suited to this journal. The article is well-written and illustrations are clear. The methodology and results presented clearly demonstrate improved performance compared with other previous/alternative methods cited (eg Sacks et al. 2010) however I am left with some significant questions

about whether they have really demonstrated improved results over generic guesses for sowing and harvesting dates. I shall attempt to state these issues below:

1. I think their method for estimating monsoon rice sowing date across time and space is likely to be good, however I am less convinced with the estimate of harvest date. From what I can gather, they make the assumption that the retreat of the monsoon is predictably associated with rice harvest, with certain lag. This is very often not the case in South Asia, with the monsoon stopping well before rice harvest in many years (leaving crops with a dry finish) while in other years the crop is ready and waiting for harvest in the field with the monsoon ongoing and preventing farmers from harvesting. In my experience of working in this region with rice-wheat systems (10 years now), I have never felt that rice harvest is reliably predictably from date of monsoon retreat. They present results to show the degree to which observed harvest dates and estimated ones agree, but I believe the variability/error in that might overshadow all the improvements in estimation of sowing date they have gained?

2. I am also wondering why estimation of harvest date is required by crop models to which this method might be applied? Don't crop models actually simulate when the crop is ready for harvest, given an input of 'sowing date'? Estimating the sowing date is of great value as a model input parameter, no doubt, but why estimate harvest? Why not simulate?

3. Even if not simulating harvest date, wouldn't it be better to simply use their climate models to accumulate daily thermal time after sowing to better estimate harvest time, rather than using retreat of the monsoon?

4. I also understand that the authors have used a fixed time (days) after retreat of the monsoon to estimate wheat sowing date. This is similarly worrying for me, due to the variability in when the monsoon retreats, compared with the actual year-to-year constancy of the date in which many farmers (particularly those with access to irrigation water for growing rabi wheat) sow their wheat crops in South Asia. Sowing of rabi wheat

is almost never triggered by soil water availability or time from the monsoon finish, or rainfall. It is triggered by a recommended ideal or optimum sowing date, advised by the local agronomic extension service or university. When you have irrigation water, you are not dependant on rainfall for sowing. All rabi wheat in South Asia is irrigated to the best of my knowledge. So, once again I would suggest that the optimum sowing time for rabi crops like wheat is a better estimation of actual wheat sowing date than the method the authors have presented. This will vary slightly between locations in South Asia, but not greatly. I suspect a lot less than the estimated sowing dates from the author's methods. Also, this 'optimum' sowing date will change with a changing climate of course, but crop models can simulate that.

5. Once a sowing date for wheat has been estimated (by whatever method), I would suggest that the best method to estimate harvest date is simply to simulate it using crop models. They take into account thermal-time accumulation requirements for different crops and varieties, slowed or hastened by things like water stress, N-stress etc.. But if an even more simplified method is desired just from climatic data, why not just accumulate thermal time from the climate data following the estimated sowing of the crop, until the specified thermal time requirement for that crop is met?

6. Lastly, I guess a significant reason why I am not yet convinced by the presented methodology is that the authors have not provided adequate evidence that it works for estimating things like "crop duration" which eventually leads to crop yield (most likely the key aspect on which an adaptation strategy is assessed). Just showing error in sowing date and harvest date and claiming that they're individually not too bad doesn't fill me with confidence. For example, a 15% +ve error in estimating sowing date, combined with a 15% -ve error in harvest date could mean a 30% error in crop duration. This would have a huge effect on gran yield. Apart from estimating sowing date correctly, how often does method estimate 'crop duration' correctly? That would be more meaningful question for me. A check on whether their method is regularly getting 'crop duration' correct would be a good test that they may be easily able to add

to this manuscript?

Forgive me if I have misunderstood aspects of the paper, as I am not a climatologist, I am an agronomist, however I suggest that the authors need to respond to these points adequately and explain why their method is better than doing what I have suggested, before their work is suitable for publication. I like the basic premise of their work, and I congratulate them on it. But I think it could be better, and could be evaluated more robustly.

Other miscellaneous points- Title – wouldn't 'Estimating' or 'Predicting' be a better word than 'Defining' in this context? Ln 12 – replace 'are' with 'is'; insert 'more representative' before 'climate' Ln 24 – replace 'site' with 'field' ? Ln 61 – add 'and cropping environments (soils etc)' after 'climatic conditions' Ln 89 – Basmati rice is really only grown in Pakistan to my knowledge, but most 'local' rice varieties are long-season and highly photo-period sensitive. Ln 307 – 'is still good agreement' – this is very qualitative. Why is it 'good'? What are your criteria for 'goodness'? Ln 403 – 'Sowing and harvest dates are an important input within crop models...' - is this true? Sowing date is an important input, yes, but harvest dates are usually simulated??? What crop models are you talking about that need to be told the harvest date?

---

## Author Comment (AC1) · 22 Jan 2018

**Authors response to reviewer comments on Manuscript**

Camilla Mathison, Chetan Deva, Pete Falloon and Andrew J Challinor

January 19, 2018

Thank you to both reviewers for their constructive and detailed responses to our manuscript. We have responded to each of the comments below. The reviewer comments are in bold and our responses are in normal type. Section 1 contains responses to comments from the first reviewer and Section 2 to comments from the second reviewer.

**1    Reviewer One**

- **What constitutes a reasonable amount of time between planting and harvest, as derived using model simulations and in light of the need to better represent these rice-wheat systems? The authors might briefly include measures or notes of what constitutes a reasonable representation in the Methods section and apply these in the Results.**

  We will add text to the methods section to make clearer what constitutes a good or reasonable representation and update the results to reflect the additional information.

- **Caveat that these methods demand a good representation of the monsoon system and that the limitations in applying this to the wider range of climate models which have issues with simulating the monsoon.**

  We will add text to explain that this method requires a good representation of the monsoon and that this presents a challenge for some climate models although not for the precipitation observations which are also used to demonstrate the method.

**Response to specific reviewer comments:**

- **Line 111: Discussion of Sacks and potentially cutting down this section**

We propose removing Fig 1 and just highlight the problems with Sacks using the text and Fig 2. This should enable the first paragraph of this subsection to be trimmed down without losing the meaning and the motivation for the paper.

- **Section 2.2: Include a bit more discussion on why this particular index was selected.**

  - **How does the NPPI provide that advantage to you over, say, another metric and/or threshold (Zeng et al. 2004; Li and Zhang 2009; Saini et al. 2011; Dong et al. 2015; Guimberteau et al. 2012; Zhang; Wang and Lin 2002; Goswami et al. 2006)?**
  - **Were other indices considered and discarded? If so, why? The authors mention a few others in the beginning of the section but do not motivate the NPPI over the others, aside from saying what it is.**

NPPI was selected because it had been used previously for analysis of regional climate models of a similar resolution. We also tried using the approach described in Sperber et al. (2013) which closely follows that from Wang and LinHo (2002). Sperber et al. (2013) defines monsoon onset as the pentad where the relative rainfall rate (relative to January) exceeds 5 $mmday^{-1}$ during May-September. However Sperber et al. (2013) regrids to GPCP data which is much lower resolution than the 25km resolution data we are using here. In the NPPI method the only regridding that takes place is to ensure that the observations and the model are on the same grid - they are both 25km so there is no loss of resolution in doing this. The threshold for NPPI is also independant of the resolution of the data which is not the case for Sperber et al. (2013).

We also looked at some more agricultural specific definitions of monsoon onset and retreat that included breaks in the monsoon, as the germination of crops can be extremely sensitive to dry periods of more than 10 days. The model data are generally too noisy to estimate the monsoon statistics year on year and therefore typically estimates of monsoon statistics are calculated using a climatology (which is a long term average from 1990-2017) of precipitation. Unfortunately applying these more crop specific metrics to a precipitation climatology rather than a yearly estimate does not give great results, this is probably because the breaks that occur in a monsoon are quite variable from year to year and are smoothed out within the climatology.

We will add some more text to explain this to this section of the methodology.

- **Moving Section 3.1 to the methodology section**

  We will move this section to the methodology section.

- **Lastly, Lucas-Picher (2011) note on their page 859 that the NPPI was used specifically as a standard with which to compare models, and that these periods may not specifically represent the beginning or end of the actual monsoon. Additionally and incidentally, they also note large range in NPPI across models, which led them to conclude that key physical monsoon processes may be missing from many model environments. How do the authors interpret these issues for their work here? Perhaps a statement should be added about the careful need to use models that are most representative of monsoon processes (not just that RCMs are better), and what the best measures are to evaluate this.**

  In the same way Lucas-Picher et al. (2011) uses the 1981-2000 climatology we use a 1990-2017 climatology. Therefore the onset and withdrawal correspond to an index computed from the precipitation climatology and normalizes the values of the pentad to generate a value between 0 and 1 which removes the bias of the model computation. Although the NPPI does not correspond to an actual date for onset and retreat that can be compared for a particular year it provides the first pentad at which the index exceeds the onset threshold of 0.618 and the first pentad after onset where the index falls below 0.618 again for the climatological period. It also provides the pentad during which the index is equal to 1.0 which represents the peak of the monsoon. The index therefore provides a pentad of onset and retreat for the climatological period. The pentad is then used to find the 5-day window for the climatological period where onset and retreat occurs which is comparable to observations. We will clarify this in Section 2.2 of the paper.

  These RCMs and in fact the driving GCMs were specifically selected because they are able to capture the precipitation of the ASM. If the driving model does not capture the ASM the method would be less effective so this is a pretty fundamental requirement. The latest generation of models capture the winds much more closely than these AR4 models but their precipitation is too low which makes them less useful in understanding the monsoon. The AR4 models we use here provide a better representation of the precipitation than the majority of the newer generation of models, so although the winds are not captured as well and they are missing potentially influential processes that are available in newer models, the precipitation is captured which is very important for agricultural applications. In general the monsoon is a

challenge for models because of its variability and complexity, there is much more that needs to be done to improve the representation of the monsoon in climate models. This method will become more robust with improving representations of the monsoon in climate models. We will add text to make these limitations clearer in the manuscript.

- **Section 2.3.1 Line 204: I may just be unable to find this information, but how was the "area averaging" applied to the RelMonsooncroprules? While the gridbox heterogeneity may be incorporated into the initial calculations of RelMonsooncroprule (as the authors mention in line 217), I wonder how the area-averaging actually impacts the authors goal of capturing some of the spatial variation, which to me seemed a big goal of the study.**

  The croprule itself is calculated on a gridbox by gridbox basis and then an area weighted area averaging is applied afterwards to obtain a single value for the whole of South Asia. The application of a gridbox by gridbox croprule, while it produces perfect results for the present day is of limited use over and above the observations because this just makes the monsoon statistic look exactly the same as the observations. In addition it limits the usefulness of the method in regions where there is not a good coverage of observations. The gridbox heterogeneity is provided by the estimate of the monsoon statistic being used which is allowed to vary between gridboxes.

  We will add text to make this clear.

- **Section 2.3.1 Line 222-225: I would be careful to caveat this statement in relating it to the ASM, as the authors found that the methods needed to be adjusted for the northeast monsoon period in southern India (which will also affect Sri Lanka as well).**

  We will add text to make this clear and modify the manuscript to say something like: 'On the basis that most of the South Asia region is dominated by the ASM, the $RelMonsoon_{croprule}$', though tuned using India observations, can be applied to regions of South Asia that are dominated by the ASM in order to estimate sowing and harvest dates for larger areas with a rice-wheat rotation. The methodology does not currently perform as well for parts of southern India where the climate is also influenced by the Northeast Monsoon but could be modified to provide better results for these areas.'

- **Figure 3: It would be helpful in Figure 3 to put NPPI calculation into box 2.3 and $RelMonsoon_{croprule}$ /MonsoonDerived-CropDate into box 2.4, to make those linkages explicit.**

We will amend this figure to reflect these comments.

- **Section 2.4 Line 229-231: This SWLs versus time period is a bit confusing given that regional temperature can exceed the SWLs, as noted. Perhaps it would be less confusing if the authors just picked one type of scenario, based either on time period or SWL.**

As this paper was funded by the HELIX project which focussed on SWLs in its analysis we felt some reference to the SWLS was needed but the general use by many people of time periods makes the time periods easier to understand. The use of time periods is much more common than SWLs, however SWLS enable the analysis to focus less on the climate scenarios used and more on what the world will look like at 2, 4 or 6°. This will differ depending on when the threshold is passed. This is a benefit as it means that the new scenarios that are developed as part of new model intercomparison projects can be compared against older ones from previous projects. These older scenarios may not contain the most up to date information on populations or economics but in reality the uncertainties are still large and these older scenarios are therefore no less likely than the newer scenarios that are now available. These models simulations exceeded only the lowest SWL and this was approximately mid-century which was why we refer to both SWLs and time periods. We will explain this in Section 2.4 in order to make this section clearer and less confusing. We will also include the reference for the SWLs by Gohar et al. (2017).

- **Section 3.1 Line 261: How are you judging "compare well with observations?" Might a skill score or some other quantitative metric be included here? In fact, there are a couple of areas in the manuscript (e.g. Line 269 comparing the monsoon onset with the sowing date) where it is stated that the results look reasonable and compare well, but there is not much context given to what is meant by this.**

If the pentads of the monsoon onset and retreat were consistent with the MinAg sowing and harvest dates i.e. for rice sowing this means where the monsoon onset range was within the observation range, this constituted a good comparison. Monsoon onset is well aligned with rice sowing so there was less need for a large crop rule to shift it nearer to the sowing date. For rice harvest and wheat sowing/harvest where the differences between the observed sowing and harvest dates and monsoon metrics were larger (see Table 2 in the paper), if there was a consistent difference across the region, this also constituted a good comparison (see appendix figures C1, C2 and C3).

When comparing the MinAg sowing and harvest dates with the monsoon derived ones we use the observation uncertainty as a guide for how good the method is. If the differences were within approximately 15 days this was as good as the MinAg observations and the newly derived dates were said to 'compare well' with observations.

We will clarify this in the comparison with the APHRODITE observations and the comparison of of MinAg sowing and harvest dates with the onset and retreat and try to define good, fair, poor more clearly.

- **I recommend moving Figure B1 to Figure 4, as it's a bit difficult to distinguish between the colors and having the differences with Aphrodite on hand would be more useful and quicker to interpret. Leave Aphrodite in the absolute colorbar however!**

  We did think there is a benefit to both of these figures which was why we put one in the appendix. We chose the current Fig.4 because we mention this plot when referring to the differing characteristics of the ASM in the southern part of India and this is more obvious in Fig. 4 than in B1. However given this feed back we will either put both in the main text or swap Figure 4 and B1 around.

- **Figure 5: This is showing a difference in days, correct? So sowing in the lightest blue areas in the northwest of India are 15 days ahead of the onset? If this is the case, then the colorbar title should be changed to 'difference in days' or similar, not 'Day of Year'.**

  Yes the blue areas represent the regions where the sowing is ahead of onset or equivalently the monsoon onset occurs after sowing and this figure shows the 'difference in days of year' so the title of the colourbar will be modified to reflect this.

- **Section 3.2 Line 277: Table 1 seems to suggest that rice is sown 19 days before the onset as calculated using the Aphrodite dataset, if I'm reading that correctly (or at least there's an 19 day gap between onset and sowing). How does this align with the comments in the introduction (Line 84), which said that rice is usually sown with the first rains? I ask because this is also relevant to judging how "well" the onset and sowing periods overlap, and can be used to assess how well the method works. If the cropping systems are so tightly scheduled as they are described in the Introduction (Line 90-97), then this difference between sowing and onset could be impactful, no? What constitutes too much of a gap between sowing and onset?**

This comment and reply links closely to the comments on section 3.1 of the manuscript above. In general we do not expect the monsoon statistics to be exactly the same as the sowing and harvest dates, this is the reason we introduced a crop rule to move the monsoon statistics to more closely reflect the observed sowing and harvest dates. Rather this method relies on the fact that in general there is consistency between the monsoon and the sowing and harvest dates across the region. This means that even if the difference between the monsoon onset and the sowing or harvest date is large this difference is similar across India. This means that wheat sowing is about the same amount of time from monsoon retreat across India and wheat harvest is about the same amount of time from monsoon onset. Although these sowing and harvest events may not be dictated by the monsoon there is a consistency there in the crop practises that we can use to estimate the sowing and harvest events through the year.

As you go from west to east across the Indo-Gangetic Plain (IGP) there are differences in crop practices. In the eastern IGP, where the farms are often small and poorer, there is a tendancy to leave sowing until the monsoon is well established to avoid crop failure that can occur if there are any long breaks in the monsoon. In these regions there is also a tendancy for lower yields particularly of wheat because the late sowing of rice leads to a delay in sowing of wheat. The wheat season, already quite short for the eastern IGP is therefore made even more so. This is mentioned around line 88.

In the introduction we will link the paragraphs between line 81 and 95 more clearly so that it is clear that although Kharif crops are traditionally sown with the first monsoon rains there are local variations, this should link more clearly with the mention of the IGP. In the methods section we will add some text to explain more clearly that the method relies on the consistency of the sowing and harvest dates with the onset and retreat rather than matching the physical dates exactly.

- **And along these lines, I'm a bit curious about the ERAint and ECHAM5 results (Looking at Table 1 and Figure 6, for example). With respect to rice sowing, the India average results are much closer to the onset it seems (3.4 and 10.1, respectively). How do we qualitatively compare these results across models and observations?**

Table 1 is an average for all South Asia which includes some areas that have very different values to the rest of the region so this will have some impact on the values in the table. The aim of showing the different models is to show that while there are some differences there is still quite a lot of consistency between them. The use of the APHRODITE

precipitation observations is intended to show that the models are not too far from reality. We would not expect all the different datasets to give the same results. The fact that APHRODITE requires the application of a larger crop rule value to the monsoon onset indicates that the APHRODITE data has a later monsoon than the models.

We will add some text to this effect to Section 3.2.

- **In Figure 6 anyway, the coverage of blue makes ECHAM5 look - dare I say it better than Aphrodite?! This is of course assuming that the Bodh et al dataset tells us something physical about the monsoon onset, in that farmers plant very close to or upon the onset (and thus sowing is a good proxy for the onset). A little guidance on how to interpret this would be appreciated, particularly with respect to the Aphrodite results (which I presume would be the most realistic?).**

Figure 6 is intended to highlight that the rice sowing dates are really closely aligned with the monsoon onset in the different datasets we have available for this analysis. It is difficult to say where one of these datasets is better than another. Generally across India the monsoon onset is within the range of observed sowing and harvest dates which means they provide a good approximation to the sowing and harvest dates, however where there are red and yellow regions in all of these plots (indicating that the monsoon statistic range of days misses or only overlaps the observation range of sowing days) these areas are different between the models which makes it difficult to say where one is better than the other. We agree though that ECHAM5 appears to have the smallest total area of red/yellow regions which does imply that ECHAM5 onset provides the closest approximation to the sowing dates for rice, this is most likely because ECHAM5 usually has an earlier onset than the other datasets which makes it closer to the rice sowing dates.

We will add some text to explain this in Section 3.2 in addition to the changes suggested below to the explanation of Figure 6 and the colourbar.

- **As an aside, it can be a bit confusing over the course of reading the paper to distinguish "before" or "after" relative to the monsoon statistics and sowing/harvesting. It's just a bit hard to keep the differencing straight. If there's a way to clearly and consistently indicate sowing dates that are before or after the onset, for example by scaling or color coding them explicitly, that would be very helpful.**

We will modify Table 1 to remove the minus signs. The table cannot include colours but we can make the text bold for before and normal type for after to make it clearer where the sowing/harvest is before/after the monsoon statistic. For simplicity in the calculation we always subtract the observations from the monsoon statistic to get the crop rule. This means that where the sowing/harvest dates are after the monsoon statistic we add the crop rule by virtue of subtracting a negative crop rule, this was why the minus signs were included in the original table.

- **Figure 6: This figure has potential to be very helpful, particularly related to my comments above, but is confusing in its current form and description. I think "hit","overlap" and "miss" need to be more clearly defined, as there's an "overlap" color and the bar itself is called "overlap". The text, Lines 270-275, do not provide any further clarity in their current form, as both the word "around" and "at least close to" are used for the blue and the yellow. These are too vague to be helpful in interpreting the maps.**

We will modify the colourbar title to say: 'Proximity of the monsoon onset in days of year to the range of days of year for observed rice sowing' We will modify the description of Fig. 6 from line 270-273 which currently reads:

Figure 6 shows the comparison between the rice sowing MinAg observations compared with the monsoon onset in the simulations and APHRODITE observations; the blue regions show that rice sowing occurs around the time of monsoon onset for a large proportion of India with the model within or at least close to (yellow regions) the range of the observations.

to say the following:

The monsoon onset and retreat estimates are provided in days of year with an uncertainty of plus or minus 2.5 days. The MinAg observations are also provided in ranges of days of year with a typical range being approximately 15-days. Figure 6 is designed to summarize how well aligned the monsoon onset range is to the observed range of rice sowing dates i.e. how the 5-day onset windows coincide with the 15-day sowing window. If the monsoon onset range is completely within the range of sowing days provided by the observations then this is classed as a 'hit' in Fig. 6 (shown by the blue regions). If the monsoon onset range is completely outside the range of the observed sowing days then this is classed as a 'miss' in Figure 6 (shown by the red regions). The yellow regions in Fig. 6 show the places where the monsoon onset overlaps the range of observed sowing days but does not completely fall within it; these regions are labelled 'Overlaps'.

**Technical Comments:**

- **While I did not catch many technical faults with the manuscript, I would say in general, I the Intro and Motivation could be condensed a bit. It reads a bit long now and some of the material can be more succinctly stated.**

  We will look into this, the removal of Fig. 1 and the subsequent redrafting of the introduction/motivation to accomodate this will address these changes.

- **Also Section 2.3.1 may not be needed as you move directly to Section 2.4**

  We are not sure we follow this. We think we do need section 2.3.1 as this is the detail of the method - i.e how we get from the monsoon statistics to the new sowing and harvest dates derived from the monsoon? We have assumed the reviewer was referring to the earlier recommendation to move section 3.1 to section 2. If we have misunderstood please let us know, otherwise we will move section 3.1 to section 2.

- **The Discussion and Conclusions section can be condensed I think, as the beginning of the Discussion section (Lines 335-343) already read like the beginning to a conclusion.**

  We will revisit these sections and see if there are areas we can reduce the text.

- **I would lastly request that the authors check the titles of all their colorbars, as some (like Figure 5) I don't think quite convey the intent of the figure.**

  We will check all the plots and the colourbars and correct any that are unclear.

**2 Reviewer Two**

**2.1 Main review comments**

1. **I think their method for estimating monsoon rice sowing date across time and space is likely to be good, however I am less convinced with the estimate of harvest date. From what I can gather, they make the assumption that the retreat of the monsoon is predictably associated with rice harvest, with certain lag. This is very often not the case in South Asia, with the monsoon stopping well before rice harvest in many years (leaving crops with a dry finish) while in other years the crop**

**is ready and waiting for harvest in the field with the monsoon ongoing and preventing farmers from harvesting. In my experience of working in this region with rice-wheat systems (10 years now), I have never felt that rice harvest is reliably predictably from date of monsoon retreat. They present results to show the degree to which observed harvest dates and estimated ones agree, but I believe the variability/error in that might overshadow all the improvements in estimation of sowing date they have gained?**

We agree with the reviewer that the method is the strongest in estimating the rice sowing date and the rice harvest and its association with the monsoon retreat is not as strong. The large variability of the monsoon phenomena is often the challenge for the modelling community. In this paper we use a climatological estimate of the monsoon onset and retreat, in this case a 17-year period. This approach is used because the data are too noisy to calculate the monsoon statistics on a year by year basis. As a result of this approach the large variability in the monsoon on a year by year basis is smoothed out and as a result highlights a consistency between the sowing/harvest dates and the climatological monsoon statistics. It is this consistency between the monsoon statistics and the crop dates that we have tried to exploit in developing this method. We focus on when the crops are planted and harvested (rather than why) and demonstrate that there is an empiracle relationship between the monsoon statistics of onset and retreat that is consistent with the sowing and harvest dates. We will make this clearer in the text and note in the introduction that this method is aimed at large scale modelling not advising farmers.

2. **I am also wondering why estimation of harvest date is required by crop models to which this method might be applied? Don't crop models actually simulate when the crop is ready for harvest, given an input of sowing date? Estimating the sowing date is of great value as a model input parameter, no doubt, but why estimate harvest? Why not simulate?**

This work has been done as preparation for the generation of ancillary files for the JULES impacts model, the crop model in JULES requires a thermal time ancillary for each crop being simulated (Osborne et al., 2014). In this situation, in order to generate the thermal time ancillary an estimate of both the sowing and harvest dates for the crops being simulated is needed. Many other crop models, for example GLAM (Challinor et al., 2004), as the reviewer points out do calculate the thermal time using temperature and the cardinal temperatures. Using a crop model may be the preferred option, however setting up a

crop model is a complex problem requiring lots of data which is often not available and generally involves a lengthy tuning process. Part of the novelty of this method is that compared to the complexity of running a crop model it does not require large amounts of data and it also is relatively simple to do. We will add discussion of this to the manuscript.

3. **Even if not simulating harvest date, wouldn't it be better to simply use their climate models to accumulate daily thermal time after sowing to better estimate harvest time, rather than using retreat of the monsoon?**

   The method presented in this paper could be used to generate sowing and harvest dates or just sowing dates depending on the requirements of the crop model being set up. If the crop model requires a thermal time the method presented provides the sowing and harvest information to generate the thermal time, however if only a sowing date is needed the user need only take this information from the output. Where a sowing and harvest date is used it is assumed that the crop model takes the thermal time as an input and uses this to ensure the crop develops throughout the crop season and is harvested at the right time. We will add text to explain this to the paper.

4. **I also understand that the authors have used a fixed time (days) after retreat of the monsoon to estimate wheat sowing date. This is similarly worrying for me, due to the variability in when the monsoon retreats, compared with the actual year-to-year constancy of the date in which many farmers (particularly those with access to irrigation water for growing rabi wheat) sow their wheat crops in South Asia. Sowing of rabi wheat is almost never triggered by soil water availability or time from the monsoon finish, or rainfall. It is triggered by a recommended ideal or optimum sowing date, advised by the local agronomic extension service or university. When you have irrigation water, you are not dependant on rainfall for sowing. All rabi wheat in South Asia is irrigated to the best of my knowledge. So, once again I would suggest that the optimum sowing time for rabi crops like wheat is a better estimation of actual wheat sowing date than the method the authors have presented. This will vary slightly between locations in South Asia, but not greatly. I suspect a lot less than the estimated sowing dates from the author's methods. Also, this 'optimum' sowing date will change with a changing climate of course, but crop models can simulate that.**

   We agree that the sowing and harvest of wheat is less strongly associated with the monsoon than rice sowing and monsoon retreat is more uncertain and variable between years. However as mentioned above using the climatological estimates of the monsoon statistics smooths out this variability and makes the consistency between the monsoon statistics and sowing and harvest dates for wheat more apparent. Even though the actual number of days between the observed sowing and harvest dates and the monsoon is quite large the amount of time across South Asia between the monsoon statistics and the rice harvest/wheat sowing is actually quite consistent between model runs. It is this consistency we are attempting to use to calculate the new sowing and harvest dates rather than a physical basis with the idea being that the monsoon provides the broader seasonality associated with different crop seasons in this region. We will add text to the methods making clear the aims of the method and include discussion of the potential limitations of the method raised by the reviewer.

5. **Once a sowing date for wheat has been estimated (by whatever method), I would suggest that the best method to estimate harvest date is simply to simulate it using crop models. They take into account thermal-time accumulation requirements for different crops and varieties, slowed or hastened by things like water stress, N-stress etc.. But if an even more simplified method is desired just from climatic data, why not just accumulate thermal time from the climate data following the estimated sowing of the crop, until the specified thermal time requirement for that crop is met?**

   As mentioned above in some cases both sowing and harvest dates are needed and using a crop model is a much more complicated option than the method presented here requiring a lot more data. However we can see a real benefit of taking the sowing date and adding an estimate of the thermal time and/or crop duration to it for each crop and comparing this with the observed harvest date. We will add this sort of calculation to the manuscript for some example varieties of rice and wheat grown in this region, which varieties will depend on the data available.

6. **Lastly, I guess a significant reason why I am not yet convinced by the presented methodology is that the authors have not provided adequate evidence that it works for estimating things like crop duration which eventually leads to crop yield (most likely the key aspect on which an adaptation strategy is assessed). Just showing error in sowing date and harvest date and claiming that they're individually not too bad doesn't fill me with confidence. For example, a 15% +ve error in**

**estimating sowing date, combined with a 15% -ve error in harvest date could mean a 30% error in crop duration. This would have a huge effect on gran yield. Apart from estimating sowing date correctly, how often does method estimate 'crop duration' correctly? That would be more meaningful question for me. A check on whether their method is regularly getting 'crop duration' correct would be a good test that they may be easily able to add to this manuscript?**

We agree that the inclusion of a comparison of the observed crop duration and the crop duration based on the estimated monsoon sowing and harvest dates would be a good test of the method. We will add this comparison to the manuscript. The observed sowing and harvest dates are given as a range of days and to do the calculations we use the midpoint. Therefore the comparison between the observed crop duration will include an estimate of the uncertainty in the range of days of sowing and harvest dates in order to compare against the estimated crop duration. Including this comparison of crop durations together with a comparison between the harvest dates from the main method presented in the paper, the thermal time approach (see point 5.) and the observed harvest dates should provide some clear evidence one way or the other on how good the outputs are from the main method presented in the paper.

**2.2 Miscellaneous points**

- **Title - wouldn't 'Estimating' or 'Predicting' be a better word than 'Defining' in this context?**

  Any of these would also be appropriate. We chose to use 'Defining sowing and harvest dates...' because we are using the estimated sowing and harvest dates in a modelling context. If the editor is happy with a change to the title we are happy to change it to 'Estimating sowing and harvest dates based on the Asian Summer Monsoon'

- **Ln 12 - replace 'are' with 'is' and insert 'more representative' before 'climate'** We will change line 11-12 to the following: The aim of this method is to provide a more accurate alternative to the global datasets of cropping calendars than is currently available and generate more representative inputs for climate impact assessments.

- **Ln 24 - replace 'site' with 'field'?**

  We will change 'site' to be 'field' so this sentence becomes: Many crop models are developed at the field scale using site specific observations to drive models and optimize outputs.

- **Ln 61 - add 'and cropping environments (soils etc)' after 'climatic conditions'**

  We will modify line 61 to say: ...assumptions and generalizations need to be made across a region with a wide variety of climatic conditions and cropping environments (soils etc).

- **Ln 89 - Basmati rice is really only grown in Pakistan to my knowledge, but most 'local' rice varieties are long-season and highly photo-period sensitive.**

  Erenstein & Laxmi (2008) suggested that Basmati rice varieties were popular in Haryana and across the IGP. However we did find information on Basmati cultivation in Pakistan as well so we will mention this as well in the manuscript. The paper focuses on India because the crop data used is from the India Ministry of Agriculture and Farming.

- **Ln 307 - 'is still good agreement' - this is very qualitative. Why is it 'good'? What are your criteria for 'goodness'?**

  This comment also relates to the comments from reviewer 1 given in Section 1 regarding section 3.1 and 3.2 of the manuscript. When comparing the MinAg sowing and harvest dates with the monsoon derived ones we use the observation uncertainty as a guide for how 'good' the method is. The MinAg sowing and harvest dates were given as a range of days of year generally about 15-day windows. So if the differences between the MinAg observations and the estimated sowing and harvest dates were within 15-days of each other this was considered a good estimate and the newly derived dates were said to 'compare well' with observations. We will make this clearer in the manuscript.

- **Ln 403 - 'Sowing and harvest dates are an important input within crop models...' - is this true? Sowing date is an important input, yes, but harvest dates are usually simulated? What crop models are you talking about that need to be told the harvest date?**

  See reply to the earlier comment number 2 in Section 2.1. In some crop models they take as input a thermal time and in order to calculate a thermal time an estimate of both sowing and harvest is needed.

**References**

Challinor, A., Wheeler, T., Craufurd, P., Slingo, J., & Grimes, D. (2004). Design and optimisation of a large-area process-based model for annual crops. *Agricultural and Forest Meteorology*, *124*(1), 99 – 120.

URL    http://www.sciencedirect.com/science/article/pii/
S0168192304000085

Erenstein, O., & Laxmi, V. (2008). Zero tillage impacts in india's rice-wheat
    systems: A review. *Soil and Tillage Research*, *100*(1-2), 1 – 14.
    URL    http://www.sciencedirect.com/science/article/pii/
    S0167198708000822

Gohar, L., Lowe, J., & Bernie, D. (2017). The impact of bias correction and
    model selection on passing temperature thresholds. *Journal of Geophysical
    Research: Atmospheres*, *122*(22), 12,045–12,061. 2017JD026797.
    URL http://dx.doi.org/10.1002/2017JD026797

Lucas-Picher, P., Christensen, J. H., Saeed, F., Kumar, P., Asharaf, S.,
    Ahrens, B., Wiltshire, A. J., Jacob, D., & Hagemann, S. (2011). Can
    regional climate models represent the Indian monsoon? *Journal of Hy-
    drometeorology*, *12*, 849–868.

Osborne, T., Gornall, J., Hooker, J., Williams, K., Wiltshire, A., Betts,
    R., & Wheeler, T. (2014). Jules-crop: a parametrisation of crops in the
    joint uk land environment simulator. *Geoscientific Model Development
    Discussions*, *7*(5), 6773–6809.
    URL http://www.geosci-model-dev-discuss.net/7/6773/2014/

Sperber, K. R., Annamalai, H., Kang, I.-S., Kitoh, A., Moise, A., Turner,
    A., Wang, B., & Zhou, T. (2013). The asian summer monsoon: an in-
    tercomparison of cmip5 vs. cmip3 simulations of the late 20th century.
    *Climate Dynamics*, *41*(9), 2711–2744.
    URL http://dx.doi.org/10.1007/s00382-012-1607-6

---

## Author Response (AR1)

**List of the main changes made to the manuscript**

All section numbers used here refer to the latest version of the manuscript. A point by point response to each of the reviewer comments which outlined the intended changes to the manuscript has been uploaded previously. Here we summarise the changes that have now been made and provide a marked up version of the manuscript.

Title: Modified from Defining to Estimating

**Summary:**

The main changes are to the methods and results sections. The new content is supportive of the original results and therefore have not required large changes to the content of the introduction and discussions.

**Introduction:**

- 1. Modifications to paragraph starting 'Agriculture in South Asia...' regarding the rice-wheat systems and the Kharif and Rabi seasons.
- 2. Added paragraph to Section 1.1 to clarify the need for harvest date information as well as sowing date information and help clarify the way these sowing and harvest dates are intended to be used. The aim of this paper is to provide a method for estimating the sowing/harvest dates from which the thermal times that are required to run almost every crop model can be estimated. In the absence of field data, these thermal times can be approximately derived from harvest dates because it would not make sense to use a crop model to derive the harvest date that is used to derive thermal times that are used as input to a crop model.
- 3. Removed the spatial comparison of the Sacks et al. (2010) and Bodh et al. (2015) sowing and harvest dates, this was Fig 1 although please note that this is not shown as deleted in the marked up manuscript.

**Methodology:**

- 1. Added sentence about the selection of the GCMs
- 2. Updated the flowchart to mention the RelMonsoon\_croprule and MonsoonDerivedCropDate explicitly

- 3. Added sentence to caveat that a model would need to be able to represent the monsoon for this method to work.
- 4. Updated Sect. 2.2 to include more information on the metrics considered and why NPPI was chosen
- 5. Moved the Section comparing the monsoon onset and retreat to precipitation observations into the methodology section from the beginning of the results (this is now Sect. 2.2.1). Please note that the deleted figure from this section does not seem to show up in the marked up version.
- 6. Updated Sect. 2.3 to explain how the area averaging is calculated.
- Updated Sect. 2.3.1 to include an explanation of what a good/fair or poor result means in this analysis.
- 8. Updated Sect. 2.4 to include more explanation of the SWLs used in the HELIX project and why we use them here.

**Results:**

- 1. Sect. comparing the monsoon onset and retreat to precipitation observations moved to the methodology section (this is now Sect.2.2.1)
- 2. Sect 3.1 updated to include a more comprehensive explanation of the Figure 6.
- 3. Table 1 and Table 2 values have been updated a bug was discovered in the code that wrote these values to the tables. Once this was rectified some of the values, particularly for the ERAint simulation were modified.
- 4. Updated Sect. 3.2 to include new analysis (A plot and supporting text) showing the crop duration for the monsoon derived datasets and the MinAg observations calculated for each state. Includes discussion of how well the method performs overall.

**Discussion and conclusions:**

Changes to mention new results. Although the new results do not change the main conclusions so these changes are small. Manuscript prepared for Earth Syst. Dynam. with version 2014/09/16 7.15 Copernicus papers of the LATEX class copernicus.cls. Date: 13 March 2018

**Defining Estimating** sowing and harvest dates based on the Asian Summer Monsoon**

Camilla Mathison1, Chetan Deva2, Pete Falloon1, and Andrew J Challinor2 1Met Office, FitzRoy Road, Exeter, EX1 3PB, UK 2School 
[revised manuscript text omitted]